# SURE-VQA: Systematic Understanding of Robustness Evaluation in Medical VQA Tasks

## Abstract

Vision-Language Models (VLMs) have great potential in medical tasks, like Visual Question Answering (VQA), where they could act as interactive assistants for both patients and clinicians. Yet their robustness to distribution shifts on unseen data remains a critical concern for safe deployment. Evaluating such robustness requires a controlled experimental setup that allows for systematic insights into the model's behavior. However, we demonstrate that current setups fail to offer sufficiently thorough evaluations, limiting their ability to accurately assess model robustness. To address this gap, our work introduces a novel framework, called *SURE-VQA*, centered around three key requirements to overcome the current pitfalls and systematically analyze the robustness of VLMs: 1) Since robustness on synthetic shifts does not necessarily translate to real-world shifts, robustness should be measured on real-world shifts that are inherent to the VQA data; 2) Traditional token-matching metrics often fail to capture underlying semantics, necessitating the use of large language models (LLMs) for more accurate semantic evaluation; 3) Model performance often lacks interpretability due to missing sanity baselines, thus meaningful baselines should be reported that allow assessing the multimodal impact on the VLM. To demonstrate the relevance of this framework, we conduct a study on the robustness of various Fine-Tuning methods across three medical datasets with four different types of distribution shifts. Our study reveals several important findings: 1) Sanity baselines that do not utilize image data can perform surprisingly well; 2) We confirm LoRA as the best-performing PEFT method; 3) No PEFT method consistently outperforms others in terms of robustness to shifts. Code is provided at `https://github.com/KOFRJO/sure-vqa`.

## 1 Introduction

Recent advancements in Vision-Language Models (VLMs) have seen increasing potential for application in the medical domain, with one key area being Visual Question Answering (VQA). In this task, VLMs could assist clinicians and can also function in medical chatbots for patient inquiries. Several general medical pretrained VLMs, such as LLaVA-Med Li et al. (2023) and Med-Flamingo Moor et al. (2023), have already been developed.

However, a crucial question remains: how robust are these models when faced with variations in data distribution during real-world application? Robustness of VLMs in medical VQA tasks refers to the ability of generating accurate answers despite variations in data, a concept also referred to as Domain/OoD generalization Yoon et al. (2024); Liu et al. (2021b) . The datasets used for training or fine-tuning may not fully capture the variations in real-world clinical data. As an example, Roberts et al. (2021) highlights how the urgency of the COVID-19 pandemic led to many studies utilizing datasets that insufficiently represent pediatric patients, introducing significant bias into the analyses. These shifts, whether through unseen disease variations, variations in the image acquisition, or different question subjects, may cause performance degradation. Understanding how robust VLMs are to these changes is key to ensuring their reliability in clinical environments.

Despite the importance of this research question, existing benchmarks fail to offer an adequate framework to address it effectively. While several benchmarks exist for evaluating the robustness of VLMs under artificial image or text corruptions (Zhang et al. (2024); Chen et al. (2023)), there

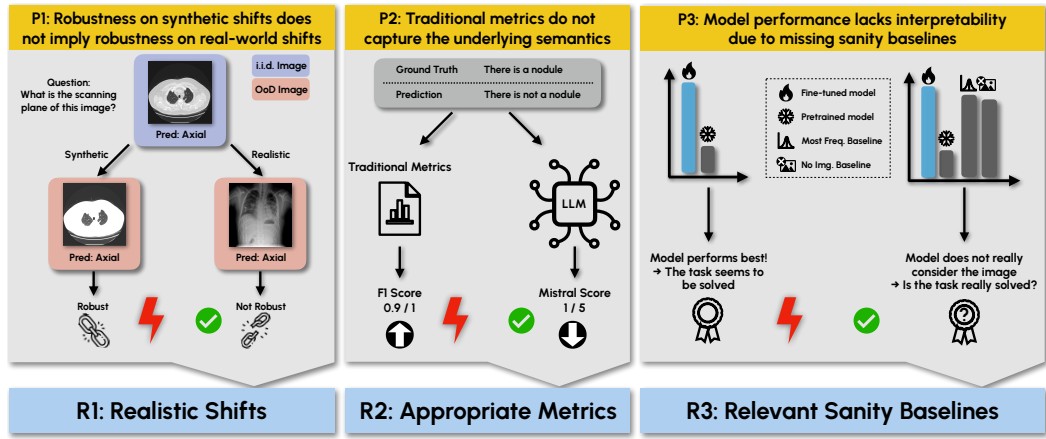

Figure 1: **Pitfalls and Requirements for Systematic Evaluating the Robustness of VLMs in VQA Tasks**. We aim to overcome pitfalls (P1-P3) in the current evaluation of VLM robustness by satisfying the three requirements (R1-R3): We define a diverse set of realistic shifts (R1). We use appropriate metrics for evaluation by using an LLM as evaluator of the VLM output (R2). Finally, we compare the results of the VLM with relevant sanity baselines to see the performance gains over such baselines like e.g. considering the text of the question only (R3).

remains a notable gap in benchmarks that account for more realistic data shifts. We address this gap in the medical domain by utilizing existing medical VQA datasets and setting them up to test VLM robustness against realistic shifts inherent to the VQA data. This focus on realistic shifts is crucial, as prior research has shown that robustness to synthetic shifts does not necessarily translate to robustness under real-world conditions (Taori et al. (2020)). Additionally, many current benchmarks rely on traditional metrics, which use token matching between the ground truth and the model's predictions. We highlight common flaws in these metrics and instead propose the use of large language models (LLMs) as evaluators, validating this approach through a human rater study. Finally, current benchmarks often overlook simple baselines, such as testing a model's ability to answer questions based solely on text. Including such sanity baselines can reveal language priors in the dataset, where questions might be easily answered either by their content or by predicting the most common answer seen during training. To overcome these three apparent pitfalls in the current literature, we define three key requirements. Based on these requirements we present a flexible framework, called *SURE-VQA*, that enables meaningful evaluation of VLM robustness in the medical domain.

To showcase the relevance of SURE-VQA, we conduct a study comparing the robustness of various fine-tuning (FT) approaches, similar to Chen et al. (2023), but with a focus on the medical domain, using LLaVA-Med as medical VLM. We focus on FT methods, including full FT and parameter-efficient FT (PEFT) because fine-tuning large VLM models is crucial and common practice for specialized tasks like medical VQA, where precision is important (Li et al. (2023); Wu et al. (2024); Singhal et al. (2023)). Our study provides valuable insights to the following questions: How do the FT methods perform in comparison to sanity baselines that, for instance, do not incorporate image content? How does the performance of different FT methods vary across medical VQA datasets? How does full FT compare to PEFT? How does the performance differ between FT methods regarding both, i.i.d. performance and robustness? Which shift is most severe regarding model robustness?

In summary, our contributions are:

1. Systematically analyze current pitfalls and based on these pitfalls formulating key requirements for meaningful robustness evaluation of VLMs in the medical domain.

2. Provide a flexible open-source framework, named *SURE-VQA*, hosted at: `https://github.com/KOFRJO/sure-vqa`.

3. Perform a human rater study confirming the importance of LLM-based metrics.

4. Show the relevance of SURE-VQA by performing a meaningful comparison of the robustness of FT methods in the medical domain leading to valuable insights for the community.

## 2 REQUIREMENTS FOR A SYSTEMATIC EVALUATION OF THE ROBUSTNESS OF VLMs

We identify several key pitfalls in the current evaluation of VLM robustness, which our proposed framework in Figure 1 addresses. In the following section, we detail these pitfalls (P1-P3) and formulate requirements (R1-R3) to overcome them. Finally, we outline the exact setup we use in our work to fulfill these requirements.

**P1: Robustness on synthetic shifts does not imply robustness on real-world shifts.**

Many existing benchmarks that focus on the robustness of VLMs, such as those in Chen et al. (2023), Shirnin et al. (2024), and Qiu et al. (2022), primarily introduce artificial perturbations to the image or text content. A notable exception is Radford et al. (2021), where the robustness of natural distribution shifts is explored in the context of their proposed CLIP model. They find that CLIP is quite effective in being more robust on natural distribution shifts of ImageNet. However, their analysis is restricted to CLIP's non-generative tasks and does not address the generative VQA task. In the medical domain, Nan et al. (2024) conduct a multimodal benchmark for evaluating robustness. Their study focuses on medical VQA tasks, but it remains limited to testing image corruptions as data shifts, leaving the crucial question of how realistic distribution shifts impact model performance open. Another study by Jensen & Plank (2022) takes a step toward addressing more realistic data shifts by examining the performance of VLMs on different versions of the VQA dataset, when training them from scratch vs. fine-tuning a pretrained model. However, their focus is primarily on linguistic variations, leaving shifts in image content under-explored. Similarly, even though the benchmark proposed by Zhang et al. (2024) uses artificial image and text corruptions, their content bias might be one step towards more realistic shifts.

However, while the robustness benchmarks in the VLM domain rarely address any realistic shifts, in the unimodal domain there is evidence that models are not robust against natural distribution shifts (Taori et al. (2020); Miller et al. (2020)). This issue has also been proven for fine-tuned, domain-specific models (Yuan et al. (2023)). Furthermore, there is evidence that artificial shifts do not necessarily translate to realistic shifts (Taori et al. (2020)).

→ *R1: Evaluate VLMs under a diverse set of realistic shifts.*

*Implementation in SURE-VQA:* We utilize three different datasets from the medical VQA domain, including SLAKE (Liu et al. (2021a)), OVQA (Huang et al. (2022)), and MIMIC-CXR-VQA (Bae et al. (2023)). On these datasets, we define several realistic shifts, spanning a range of subtleties, with some having a more pronounced impact, such as modality shifts, while others, like gender shifts, are more subtle in their effects. Furthermore, certain shifts primarily affect the image content, such as changes in the body location being imaged, whereas others influence the text input, such as shifts in question types. In an ablation, we compared the model's performance on image corruptions (e.g., blur, noise, brightness) with the realistic shifts we defined in the SLAKE dataset. The results demonstrate that artificial shifts fail to accurately capture the challenges presented by realistic shifts, supporting our argument. Further details on this study are provided in Appendix D. We also investigate the effect of multimodal shifts in comparison to unimodal ones in Appendix E. In the context of foundation models, defining i.i.d. (independent and identically distributed) and OoD (out-of-distribution) data is challenging due to the vast amount of training data. In our work, we therefore define OoD specifically as data distributions that differ from the fine-tuning data. This approach allows us to precisely control how the data distribution is modified relative to the model's fine-tuning environment.

**P2: Traditional metrics do not capture the underlying semantics.**

Many studies in the VLM field continue to rely on traditional metrics like BLEU and CIDEr (Sung et al. (2022); Chen et al. (2023); Qiu et al. (2022)), or accuracy-based metrics (Li et al. (2023); Jensen & Plank (2022); Qiu et al. (2022)), which are dependent on word- or n-gram matches. We refer to these as "traditional metrics" throughout the paper. Recent research, however, has begun to adopt a more sophisticated approach by employing LLMs to evaluate the output of VLMs (or other

LLMs) (Wang et al. (2024); Ostmeier et al. (2024); Liu et al. (2023b); Chiang & Lee (2023); Fu et al. (2024); Kocmi & Federmann (2023)).

The primary limitation of traditional metrics is their inability to capture the underlying semantics of a sentence. They fail to recognize synonyms or account for negation, often misjudging sentences that differ from the ground truth by a single token, such as "not." We illustrate examples of these failures in Appendix A.1, similar to Ostmeier et al. (2024). While we are not the first ones to propose using an LLM as an evaluator, the fact that previous studies have shown the subpar performance of traditional metrics but they are still used in many papers underlines the need to formulate it as an explicit requirement within an evaluation study for VLMs.

$\rightarrow$ *R2: Evaluate VLMs with appropriate metrics that capture the underlying semantics of the output.*

*Implementation in SURE-VQA:* We employ the Mistral model (Jiang et al. (2023)) as an evaluator, utilizing three distinct prompts tailored for different question types: open-ended, closed-ended binary, and closed-ended multilabel. To optimize computational efficiency and reduce potential errors from the LLM evaluator, we implement a hybrid metric. Specifically, when the answer exactly matches the ground truth, we assign the highest score without invoking the LLM, thereby saving computational resources and minimizing the risk of evaluation failures. Additionally, we assess the feasibility of this evaluation by conducting a human rater study, where we empirically validate its performance in comparison to traditional metrics.

**P3: Model performance lacks interpretability due to missing sanity baselines.**

Currently, the performance of VLMs is typically reported either in isolation or in comparison to other VLMs. This is evident in papers that introduce new VLMs, such as Li et al. (2023); Moor et al. (2023), as well as in benchmark studies (Chen et al. (2023); Zhang et al. (2024); Qiu et al. (2022); Nan et al. (2024)). A notable exception is the work of Liu et al. (2023a), where the performance of a language-only GPT-4 is evaluated. However, the focus here is rather to improve the model by ensembling with the LLM instead of highlighting that it might be an issue of current VQA datasets that so many questions can be answered based on the text only. Further, they do not provide a *no image* sanity baseline of their own model, leaving the multimodal usage of their own model unexplored. Another study by Parcalabescu & Frank (2023) contextualizes the multimodal use of VLMs by employing Shapley values to assess the contribution of each modality to the output.

The problem with many VQA datasets is that they tend to contain hidden patterns, allowing models to exploit shortcuts (Geirhos et al. (2020)) rather than using all available information, including the image content (Kafle & Kanan (2017); Chen et al. (2024a); Kervadec et al. (2021); Goyal et al. (2017); Dancette et al. (2021)). This means that high performance on these datasets does not guarantee that the model is actually utilizing the visual input to answer the question; instead, it might be exploiting patterns in the questions themselves (Kafle & Kanan (2017); Kervadec et al. (2021)). As demonstrated by Chen et al. (2024a), in many cases, the visual content in VQA datasets is unnecessary, and models can achieve high performance simply by relying on the textual modality. This indicates that the models are leveraging hidden biases in the question-answer pairs rather than solving the task as intended.

$\rightarrow$ *R3: Provide relevant sanity baselines to contextualize the benefits of VLM fine-tuning and multimodal information usage.*

*Implementation in SURE-VQA:* We propose that using relevant sanity baselines to reveal such dataset biases can be beneficial to explore how the models solve the given task and how the datasets are structured. Thereby, we put the performance of the fine-tuned model into context by comparing it to baselines in two aspects:

1. Not using the image information: Here, we *a)* choose the most frequent answer to the question in the training set and answer the same question in the test set with this and *b)* train the model without using any image information, which should lead to learning shortcuts based on the language.

2. Not fine-tuning the model: Use the plain VLM without any fine-tuning and report the performance on the test set. This serves at the same time as a baseline to see if or how much the shifts are inherently different between i.i.d. and OoD when not fine-tuning.

Figure 2: **Datasets and Shifts Used in the Study**. We use three datasets with four different types of shifts, resulting in seven different settings for robustness analysis. Shifts that are mainly focused on changes in the image content are shown by a change of images between i.i.d. and OoD and shifts that focus on the question content are shown by a change of the question and answer between i.i.d. and OoD shifts. The taxonomy for the shift category is partially taken from Castro et al. (2020).

# 3 FRAMEWORK SETUP

## 3.1 UTILIZED DATASETS

An overview of the utilized datasets is provided in Figure 2, with further details regarding the datasets, preprocessing steps, and split sizes available in Appendix C.3. In total, we use three different medical VQA datasets, each incorporating a variety of realistic shifts to meet the requirement outlined in R1. The taxonomy for shift categories is thereby partially taken from Castro et al. (2020), also used in related work such as Bungert et al. (2023); Roschewitz et al. (2023); Choi et al. (2023).

**SLAKE** We use the SLAKE dataset (Liu et al. (2021a)) with two different shifts: 1) *Modality shift*: Representing an acquisition shift (Castro et al. (2020)), we train the model exclusively on CT and MRI images (2D slices) and then test it on X-ray images. 2) *Question type shift*: During training, the model is exposed to questions about image content such as shape and color but excludes questions related to the size of organs. These size-related questions are introduced in the OoD test set.

**OVQA** The OVQA dataset (Huang et al. (2022)) is used with two shifts: 1) *Body part shift*: Representing a manifestation shift (Castro et al. (2020)), we train the model on images of the hand, chest, and head, and test it on images of the leg. 2) *Question type shift*: In the training set, the model is exposed to questions about various image contents like abnormalities and conditions, but questions related to the organ system are reserved for the OoD test set.

**MIMIC-CXR-VQA** We use the MIMIC-CXR-VQA (Bae et al. (2023)) dataset with three different shifts, all representing population shifts (Castro et al. (2020)): 1) *Gender shift*: The model is trained on male patients and tested on female patients. 2) *Population shift*: Training is conducted using data from white patients, with testing on patients from other ethnicities. 3) *Age shift*: The model is trained on patients over the age of 60 and tested on patients under 40. A gap is intentionally introduced between the i.i.d. and OoD groups to make the shift more explicit.

## 3.2 HUMAN RATER STUDY

**Study Design** To ensure that the scores assigned by Mistral align with human judgment, we conduct a human rater study. For each dataset, we randomly selected 50 open-ended questions where the prediction did not exactly match the ground truth, as exact matches would automatically score highest by the hybrid metric (R2). Five human raters evaluated the questions, and we calculate the

Figure 3: **Results of the Human Rater Study**. Human interrater correlation is calculated between five human raters. We use Kendall's Tau (Kendall (1945)) for calculating the correlation.

correlation between humans and Mistral's scores using Kendall's Tau (Kendall (1945)). Additionally, we report the correlation between humans and traditional metrics, and the inter-rater variability.

**Results** The results of the human rater study are presented in Figure 3, with detailed results available in Appendix B.1. Mistral demonstrates the highest correlation with human ratings on the SLAKE and OVQA datasets, with differences between Mistral and the best traditional metrics being $0.02$ and $0.13$, respectively. On the MIMIC dataset, traditional metrics perform slightly better, with the difference between Mistral and traditional metrics ranging from $0.08 - 0.18$. This variation can be attributed to several factors. In the OVQA dataset, for example, the structure of the questions and answers makes traditional metrics more prone to failure. Tokens like "fracture," "left"/"right", or specific bone names often match between the ground truth and predictions, despite significant differences in other tokens. An example of such a mismatch is illustrated in Figure 4a. On the other hand, MIMIC answers are highly structured, particularly for open-ended questions, where a fixed set of classes is listed in a comma-separated format. In these cases, traditional metrics perform well because token matching tends to be more accurate especially when the classes have distinct tokens.

Despite these limitations, Mistral's performance on the MIMIC dataset remains strong, and its correlation with human ratings is in a similar range to that observed for the SLAKE dataset. Moreover, the failures of Mistral on MIMIC are not complete failures, as shown in Figure 4b. Here, Mistral does not put out the opposite than it should but rather a wrong tendency. Further, in the first example shown on the left, also some of the traditional metrics fail. In summary, although there are instances where traditional metrics show higher correlations with human scores, Mistral proves to be generally more robust and less prone to complete failures. Its correlation remains consistently high across all datasets, confirming its suitability as a reliable metric for evaluation in VQA tasks.

## 4 EMPIRICAL STUDY ON THE ROBUSTNESS OF FINE-TUNING METHODS

To show the relevance of SURE-VQA we performed an empirical study comparing the robustness of various FT methods under realistic shift in medical VQA. This is especially important since practitioners should be informed about the differences of the FT methods not only in terms of their performance but also how robust they are when selecting a method.

### 4.1 STUDY DESIGN

We utilize (image, text) datasets from the medical domain, splitting them so that the training and testing distributions differ. As our base model, we employ LLaVA-Med 1.5, a state-of-the-art medical VLM (Li et al. (2023)). We fine-tune the model using four methods: full FT, prompt tuning (Lester et al. (2021)), LoRA (Hu et al. (2021)), and $(\text{IA})^3$(Liu et al. (2022)). Hyperparameters for the PEFT methods are selected based on the full training set and corresponding validation set for each dataset. Details regarding the hyperparameter search can be found in Appendix C.1. To measure robustness, we split the data into i.i.d. training and i.i.d. and OoD test sets, as outlined in section 3.1, thereby fulfilling R1. We then evaluate the performance of the VLM using Mistral as an evaluator, fulfilling R2. For robustness measurement, we calculate the *relative robustness (RR)* (Chen et al. (2023)), defined as $RR = 1 - \Delta P/P_I$, where $\Delta P = (P_I - P_O)$, and $P_I$ is the i.i.d test performance and $P_O$ is the OoD test performance. For a better interpretation of the results, we compare them against relevant sanity baselines as described in R3.

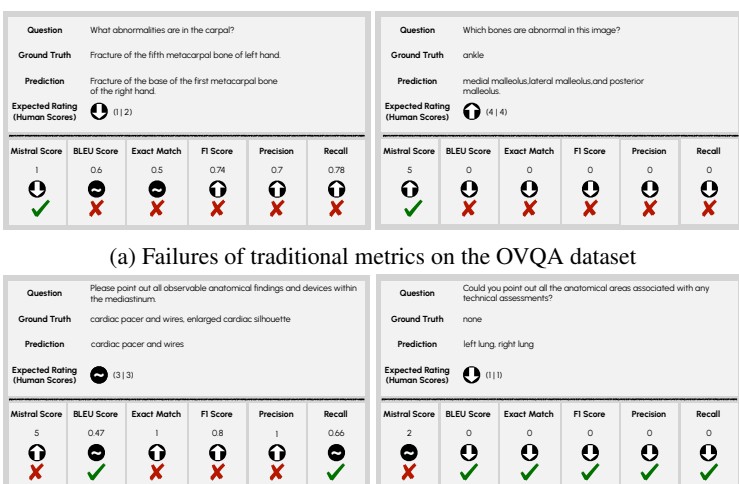

(a) Failures of traditional metrics on the OVQA dataset

(b) Failures of Mistral on the MIMIC dataset

Figure 4: **Qualitative Results of the Human Rater Study**. For each sample, the question, ground truth, prediction, and expected score by human ratings are shown. On the bottom, for each automated metric, the absolute value is shown with an indication if it is high or low in the metrics range and an indication if the expectations from the human ratings are met.

## 4.2 RESULTS

The results of the FT robustness study can be seen in Figure 5 and Figure 6. Detailed tables with the results are shown in Appendix C.4.

**Comparison to Sanity Baselines.** Generally, the PEFT models outperform the *no fine-tuned* models, the *most frequent* baseline, and their respective *no image* baselines on the i.i.d. datasets. This is in contrast to full FT, which mostly does not outperform the *most frequent* baseline. However, the gap between the *no image* baselines and the PEFT models varies across datasets. For example, it is largest on the SLAKE dataset, averaging 24%, but much smaller on the OVQA dataset at around 11%, and similarly small on the MIMIC dataset at 7%. Interestingly, the MIMIC dataset yields relatively poor results with fine-tuned models, averaging just 61.2% mistral accuracy on closed-ended questions and 3.2 Mistral score on open-ended ones on the i.i.d. set. Specifically, for prompt tuning, the performance of the model fine-tuned with images is almost identical to its *no image* counterpart, showing only about a 2% improvement. This indicates that, for the MIMIC dataset, the image encodings contribute little to the question-answering task, suggesting that the vision encoder is not able to extract meaningful information from the images. This could be because the MIMIC dataset focuses on detailed chest X-ray images and the image encoder does not have this fine-grained expertise in such a specific task. While the *no image* baseline and the *most frequent* baseline both mostly perform better than random classifier, this effect is particularly noticeable for the closed-ended questions on OVQA, where the *no image* baseline achieves on average 72%, and the *most frequent* baseline 73.6%. This suggests that the model can often rely on learned question-answer correlations without needing the images.

**Comparison Between Datasets.** For closed-ended questions, the average i.i.d. performance of the PEFT models is similar between the SLAKE (86%) and OVQA (83.5%) datasets. However, the average gap between these models and the *no image* sanity baseline is larger on SLAKE compared to OVQA as mentioned above. This suggests that image information has a greater impact on overall model performance in the SLAKE dataset. A reason for this could be that the ratio of unique questions is smaller in the OVQA dataset (see Table 13, Appendix C.3.4), and also the *most frequent* baseline is performing better on this dataset. This suggests that there are more repetitive questions and higher bias, making the model prone to shortcut learning. Continuing on closed-ended questions, the PEFT models outperform the no-finetuned model by a larger margin on OVQA, with a 41% improvement on average, compared to 29% on SLAKE. This suggests that fine-tuning has a

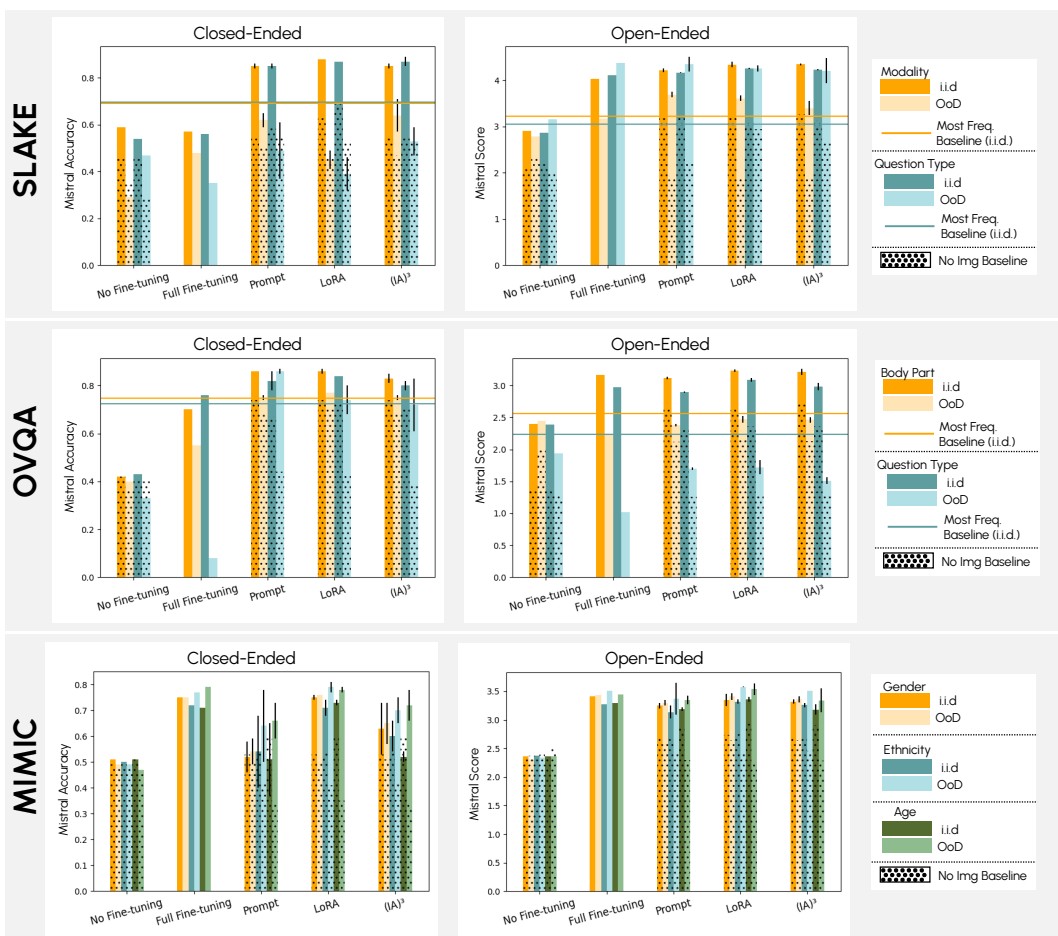

Figure 5: **Results of the FT Robustness Study on the i.i.d. and OoD Test Set**. Reported results show the mean over three seeds (exception: no FT, full FT) with the standard deviation for the non-baselines. Mistral Accuracy refers to the accuracy being rated by Mistral, meaning it assigns 0 or 1 to the output. For MIMIC, the *most frequent* sanity baseline can not be calculated as too few questions match the training set.

greater impact on the OVQA dataset. For open-ended questions, the average i.i.d. performance is the best on the SLAKE dataset with a Mistal score of 4.22. In contrast, the performance on the MIMIC dataset seems generally insufficient for practical use. For open-ended questions, the average performance improvement over the *no image* baseline across all PEFT methods is just 9%, and for closed-ended questions, it's only 5%. Especially for prompt tuning, there is a 3% decrease for closed-ended ones.

**Comparison between Full FT and PEFT Methods.** In our evaluation, full FT demonstrated inferior performance compared to PEFT methods on the SLAKE and OVQA datasets and only partially matched or slightly surpassed LoRA on some splits of the MIMIC dataset. Notably, the performance of full FT generally improved with increasing dataset size, performing worst on SLAKE, slightly better on OVQA, and best on MIMIC, aligning with observations by Dutt et al. (2024). Further, we observed a significant failure of full FT robustness for the OVQA question type shift. Overall, as full FT neither outperforms PEFT methods in terms of performance nor robustness, we agree with Dutt et al. (2024) that PEFT methods are particularly well-suited for medical and low-data scenarios.

**Comparison Between PEFT Methods.** On the i.i.d. set, LoRA is consistently the best PEFT method, achieving an average accuracy of 81.8% on closed-ended and a Mistral score of 3.6 on open-ended questions across different datasets and shifts. However, the performance differences

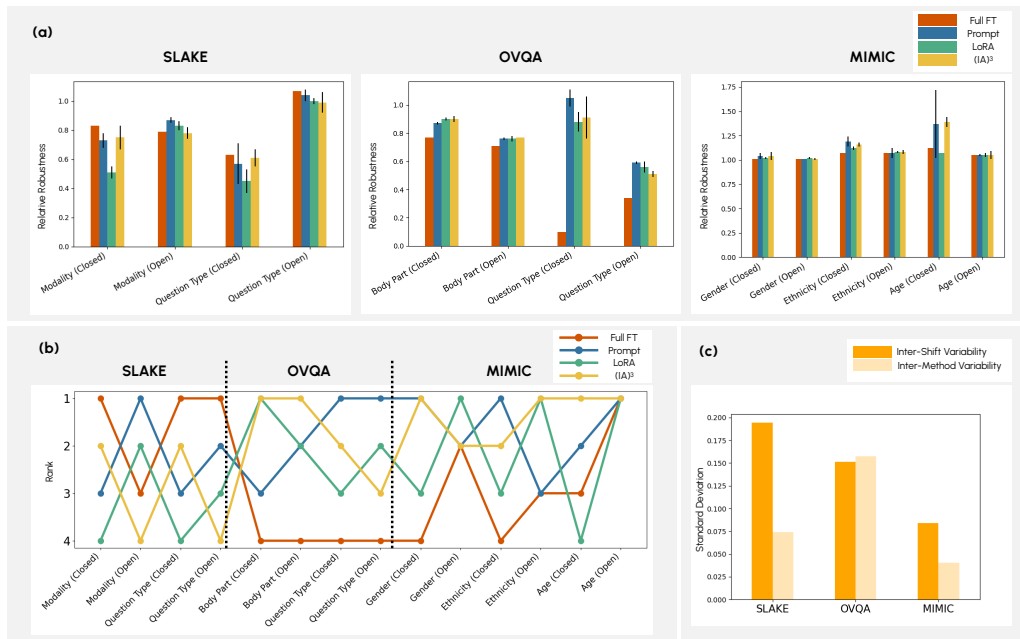

Figure 6: **Results of the FT Robustness Study Focusing on the Relative Robustness (RR)**. Since the study focuses on comparing the PEFT method and our definition of OoD holds for these methods, the *no FT* baseline is excluded. (a) RR on the three datasets for all FT methods. Results show the mean and standard deviation over three seeds (exception: full FT). (b) RR Ranking of the methods. (c) Standard deviation between shifts vs. standard deviation between FT methods.

between LoRA and other PEFT methods are quite small, especially considering that these other methods require fewer parameters to train. $(IA)^3$ performs an average of 3.5% below LoRA, while prompt tuning shows a 5% lower performance on average across the datasets. Overall, robustness within methods is more homogeneous, while there is more variance across dataset shifts. This suggests that the type of shift has a greater impact on robustness than the choice of the fine-tuning method as visualized in Figure 6c). Only on the OVQA dataset, the inter-method variability is higher than the inter-shift variability, but this is due to the failure in robustness of full FT. Only comparing the PEFT methods, also for OVQA the inter-shift variability is lower, shown in Appendix C.4, Figure 12. Besides that, none of the fine-tuning methods consistently outperforms the others in terms of robustness as seen in figure Figure 6b), where the rank of each method is depicted with respect to their RR. As one exceptional outlier of the PEFT methods, on closed-ended questions in SLAKE, LoRA demonstrates significantly lower robustness compared to other methods, with an RR of 48%, compared to 65% for prompt tuning and 68% for $(IA)^3$.

**Comparison Between Shifts.** The robustness trends vary between datasets, as shown in Figure 6a. On SLAKE, models demonstrate greater robustness on open-ended questions compared to closed-ended ones, while the opposite is true for OVQA, where models perform more robustly on closed-ended questions. This pattern is especially clear in question type shifts, where RR on SLAKE is 54% for closed-ended questions and increases to 101% for open-ended. In contrast, on OVQA, RR is 94% for closed-ended and drops to 55% for open-ended questions. At the same time, the question type shift seems most severe if the model performance drops. The differing behavior in SLAKE and OVQA datasets arises from the alignment of OoD questions with training data. While the OoD questions are not included in the training set, training questions help the model capture the necessary information to address them. This alignment is observed in SLAKE for open-ended questions and in OVQA for closed-ended ones. The population shifts on MIMIC did not affect the models' robustness, i.e. the models seem robust against such shifts since they show over 100% RR. However, since the performance on the MIMIC dataset is generally insufficient, it is questionable if this observation would hold on higher i.i.d. performance. Future experiments could investigate whether the low performance or the kind of shift is causing this behavior.

## 5 CONCLUSION AND TAKE-AWAYS

We present a framework that allows testing the robustness of VLMs in medical VQA tasks. Thereby, we especially focus on three key requirements for a meaningful evaluation of robustness.

**Empirical Confirmation of R1-R3.** While in section 2 we derived R1-R3 from flaws in the current literature, our study provides empirical evidence for their importance. **R1:** In an ablation study (Appendix D), we show that corruption shifts do not necessarily translate to real-world shifts, thereby justifying our claim to also work with more real-world shifts. **R2:** We present several critical failures of traditional token-matching metrics and prove the applicability of our LLM evaluation setup by a human rater study. **R3:** We show that some sanity baselines that do not use the image information already perform surprisingly well. This highlights two aspects: 1) As stated in R3, reporting such sanity baselines is crucial for understanding the true multi-modal performance of a VLM, beyond its language-only capabilities. 2) This observation further suggests that we need more elaborate data sets and tasks in medical VQA that minimize the potential for shortcut learning based solely on the language content. Achieving this may involve incorporating greater linguistic variability in questions, as seen in Bae et al. (2023), where questions were rephrased using GPT-4, and ensuring a broad range of semantic differences in the questions. Progress can be tracked as a low performance of the no-image sanity baseline.

**Generalizability of the Framework.** SURE-VQA, serves as a starting point for a comprehensive evaluation of robustness and it can be flexibly extended to new datasets, methods, and domains. Additionally, SURE-VQA can support method development aimed at enhancing the robustness of VLMs. In our study, we define OoD as a data shift w.r.t the FT data. However, our framework also allows to compare VLMs in a zero-shot setting without any FT, when simply re-defining OoD as a data shift w.r.t the pre-training data. However, such a definition becomes increasingly challenging to validate with foundation models, since the exact training data used is often not known. Notably, R2, and R3 go beyond robustness analysis and should be integral to any well-designed VLM study.

**Main Insights from the FT Robustness Study.** Our exemplary study which compares the robustness of various FT methods reveals several key insights. While we confirm LoRA as the best-performing FT method on the i.i.d. dataset, no single FT method consistently outperforms the others in terms of robustness. Further, in line with findings of Dutt et al. (2024), we find that PEFT methods are more efficient than full FT in the lower data regimes, especially present in the medical domain. As another finding, robustness trends appear to be more consistent within FT methods than across different dataset shifts, indicating that the type of shift has a greater impact on robustness than the choice of FT method. This suggests that robustness alone is not a decisive factor when choosing a FT method. However, the type of data shift anticipated in the test set is crucial, as different shifts may uniquely challenge model performance. Additionally, the models generally exhibit robustness against population shifts. However, further investigation is needed to determine whether this is due to already low i.i.d. performance or only because of the nature of the shift.

**Future Work.** As mentioned above, SURE-VQA is a starting point and future work can include the investigation of more datasets, shifts, methods, and models. A key research direction is the development of additional VQA datasets, particularly in the medical domain. This would not only address the need for greater diversity in question types but also improve the clinical relevance of the questions posed. While current datasets cover various question types, questions about scanning modalities, for example, may be less valuable to clinicians. More relevant questions might include for example questions related to prognosis, such as the potential spread of a tumor. We believe that collaboration with clinicians could help define and incorporate a broader set of clinically meaningful questions into future datasets. Additionally, our framework provides valuable insights into the factors affecting the robustness of VLMs. Future work could explore methods for enhancing robustness, like in Yoon et al. (2024); Ma et al. (2024). Finally, the underperformance of LLaVA-Med, one of the state-of-the-art models in medical VQA, on the MIMIC-CXR-VQA dataset indicates that there is significant room for improvement in medical VLM development. Recent work by Chen et al. (2024b) has focused on building foundation models specifically for chest X-ray data, using datasets like MIMIC-CXR-VQA. However, future efforts could aim to develop more robust models capable of handling multiple modalities across a wider range of clinical scenarios.

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

## A  EVALUATION DETAILS

### A.1  FAILURES OF TRADITIONAL METRICS

Examples of failures of traditional token-matching metrics are shown in Figure 7

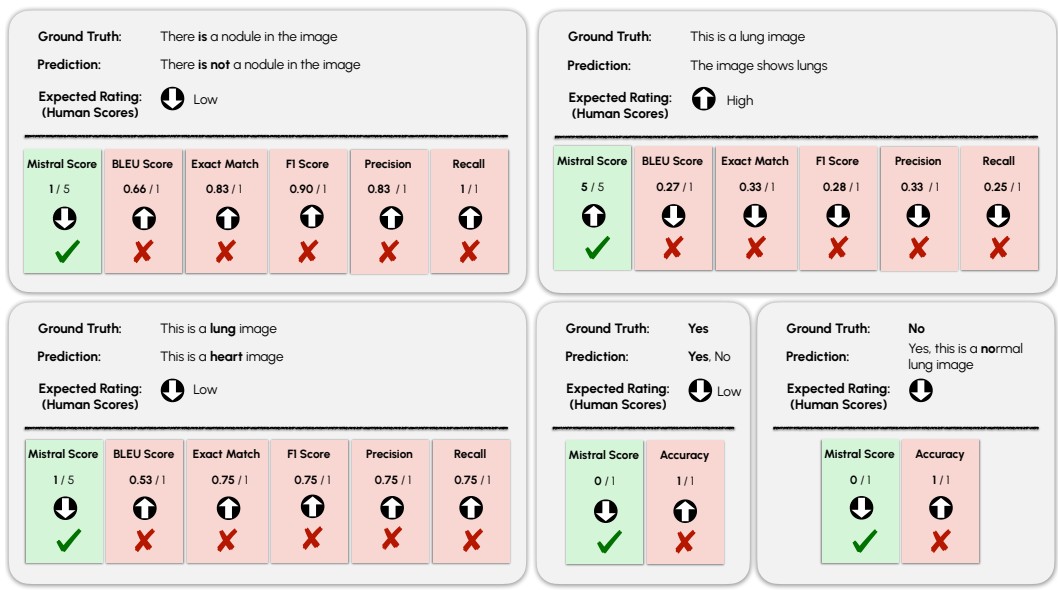

Figure 7: Failures of Traditional Metrics

## A.2 PROMPTS FOR EVALUATION

Listing 1: Mistral Prompt for Evaluating Open-Ended Questions

```
[INST] You are a helpful evaluator to evaluate answers to
    questions about biomedical images.
Score the following answer to a question about an image with
    respect to the ground truth answer with one to five stars.
Where the stars have the following meaning:
 1. One Star: "Incorrect"
    - The answer does not match the ground truth and contains
        significant inaccuracies.
    - Demonstrates a clear misunderstanding or misinterpretation
        of the question.
 2. Two Stars: "Partially Correct"
    - The answer has some elements that match the ground truth,
        but there are notable discrepancies.
    - Shows partial understanding but lacks overall accuracy in
        addressing the question.
 3. Three Stars: "Mostly Correct"
    - The answer aligns with the ground truth to a reasonable
        extent, but there are some inaccuracies or gaps.
    - Demonstrates a moderate understanding but may lack
 4. Four Stars: "Correct with Minor Deviations"
    - The answer is largely accurate and corresponds closely to
        the ground truth.
    - Minor deviations or omissions are present but do not
        significantly impact the overall correctness.
 5. Five Stars: "Perfect Match"
    - The answer exactly matches the ground truth with no
        discrepancies.
    - Demonstrates a precise and complete understanding of the
        question, providing a flawless response.
Here are some instructions on the input and output format:
 - The input will be passed as json format with the following
    fields that are important:
    - "question": the question about the image
    - "gt": the ground truth answer to the question
    - "pred": the predicted answer to the question
```

```
  - The output should be in json format and look the following:
     { mistralscore: <xxx>}
    where <xxx> is the number of stars you give to the answer.
        Do not add anything else to the answer.
  [/INST]

```

Listing 2: Mistral Prompt for Evaluating Closed-Ended Questions

```
[INST] You are a helpful evaluator to evaluate answers to
    questions about biomedical images.
Score the following answer to a question about an image with
    respect to the ground truth answer with zero or one star.
The questions are all close ended, therefore the answer is
    either correct or false, there are no states in between.
Where the stars have the following meaning:
 0. Zero Star: "Incorrect"
   - The answer does not match the ground truth and contains
       significant inaccuracies.
   - Demonstrates a clear misunderstanding or misinterpretation
       of the question.
 1. One Star: "Perfect Match"
   - The answer exactly matches the ground truth with no
       discrepancies.
   - Demonstrates a precise and complete understanding of the
       question, providing a flawless response.
Here are some instructions on the input and output format:
 - The input will be passed as json format with the following
     fields that are important:
   - "question": the question about the image
   - "gt": the ground truth answer to the question
   - "pred": the predicted answer to the question
 - The output should be in json format and look the following:
     { mistralscore: <xxx>}
    where <xxx> is the number of stars you give to the answer.
        Do not add anything else to the answer.
  [/INST]

```

Listing 3: Mistral Prompt for Evaluating Closed-Ended Multilabel Questions

```
[INST] You are a helpful evaluator to evaluate answers to
    questions about biomedical images.
Score the following answer to a question about an image with
    respect to the ground truth answer with 0, 0.5 or 1 star.
Each question asks for two options in the image and the answer
    can either be one of the options, both of the options or
    none.
The stars for rating have the following meaning:
 0 Star: "Incorrect"
   - The answer does not match the ground truth and contains
       significant inaccuracies.
   - Demonstrates a clear misunderstanding or misinterpretation
       of the question.
   - This is the case if
     - Option A is the ground truth answer, but the prediction
         is Option B
     - Option B is the ground truth answer, but the prediction
         is Option A
     - The ground truth answer is "both", but the prediction is
         "none"
     - The ground truth answer is "none", but the prediction is
         "both"
 0.5 Star: "Partially Correct"
```

```
        - The answer partially matches the ground truth, but
            contains some inaccuracies.
      - Demonstrates a partial understanding of the question,
            providing a partially correct response.
      - This is the case if
        - Option A/B is the ground truth answer, but the
            prediction is "both"
        - Option A/B is the ground truth answer, but the
            prediction is "none"
        - The ground truth is "both", but the prediction is option
            A/B
        - The ground truth in "none", but the prediction is option
            A/B
  1 Star: "Perfect Match"
      - The answer exactly matches the ground truth with no
            discrepancies.
      - Demonstrates a precise and complete understanding of the
            question, providing a flawless response.
      - This is the case if
        - Option A is the ground truth answer and the prediction
            is Option A
        - Option B is the ground truth answer and the prediction
            is Option B
        - The ground truth is "both" and the prediction is "both"
        - The ground truth is "none" and the prediction is "none"

  Especially for the "none" Cases:
      When the ground truth is "none":
          If the prediction is "none", the score should be 1 star
              .
          If the prediction is "both", the score should be 0
              stars.
          If the prediction is Option A or B, the score should be
              0.5 stars.
      When the prediction is "none":
          If the ground truth is "none", the score should be 1
              star.
          If the ground truth is "both", the score should be 0
              stars.
          If the ground truth is Option A or B, the score should
              be 0.5 stars.

  Especially for the "both" Cases:
      When the ground truth is "both":
          If the prediction is "both", the score should be 1 star
              .
          If the prediction is "none", the score should be 0
              stars.
          If the prediction is Option A or B, the score should be
              0.5 stars.
      When the prediction is "both":
          If the ground truth is "both", the score should be 1
              star.
          If the ground truth is "none", the score should be 0
              stars.
          If the ground truth is Option A or B, the score should
              be 0.5 stars.

  Here are some instructions on the input and output format:
   - The input will be passed as json format with the following
        fields that are important:
      - "question": the question about the image
      - "gt": the ground truth answer to the question
      - "pred": the predicted answer to the question
   - The output should be in json format and look the following:
```

```
        { mistralscore: <xxx>}
      where <xxx> is the number of stars you give to the answer.
          Do not add anything else to the answer.
    [/INST]
  
```

## B  HUMAN RATER STUDY DETAILS

### B.1  DETAILED RESULTS OF THE HUMAN RATER STUDY

The following figures show detailed results of the human rater study. Figure 8, Figure 9, and Figure 10 show scatter plots with the correlation between the human ratings and the other metrics. Figure 11 shows detailed correlation results, including the correlation between Mistral and the other metrics.

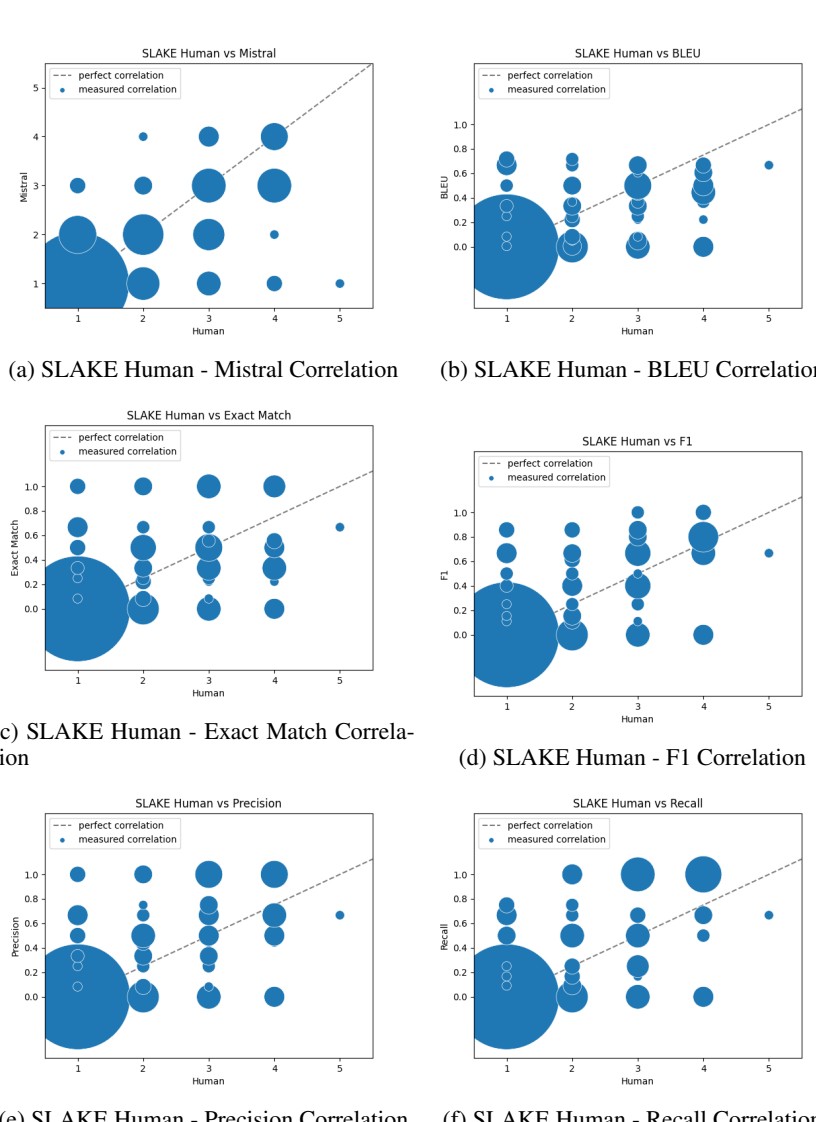

(a) SLAKE Human - Mistral Correlation

(b) SLAKE Human - BLEU Correlation

(c) SLAKE Human - Exact Match Correlation

(d) SLAKE Human - F1 Correlation

(e) SLAKE Human - Precision Correlation

(f) SLAKE Human - Recall Correlation

Figure 8: Scatter plots showing the correlation between the human ratings and respective other metrics on the SLAKE dataset. Size of the dots indicates the number of ratings that correspond to that point.

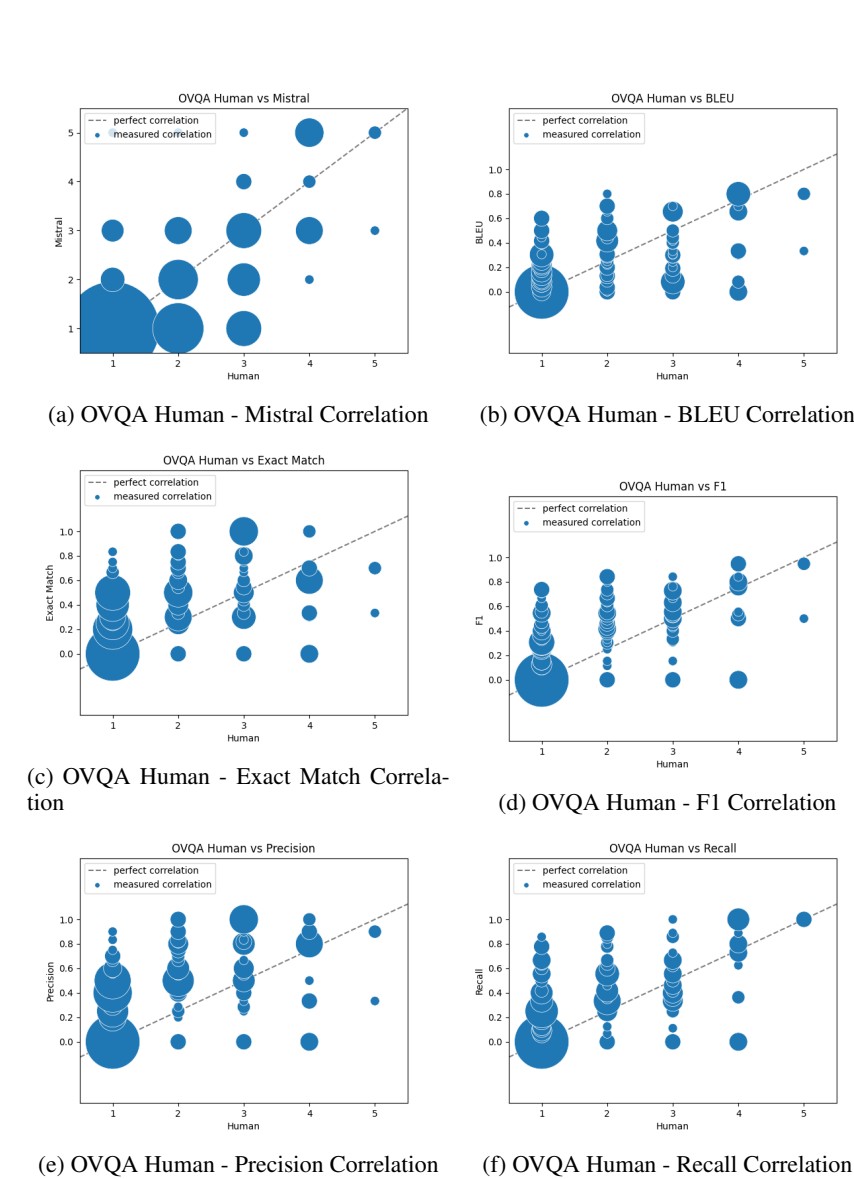

(a) OVQA Human - Mistral Correlation

(b) OVQA Human - BLEU Correlation

(c) OVQA Human - Exact Match Correlation

(d) OVQA Human - F1 Correlation

(e) OVQA Human - Precision Correlation

(f) OVQA Human - Recall Correlation

Figure 9: Scatter plots showing the correlation between the human ratings and respective other metrics on the OVQA dataset. Size of the dots indicates the number of ratings that correspond to that point.

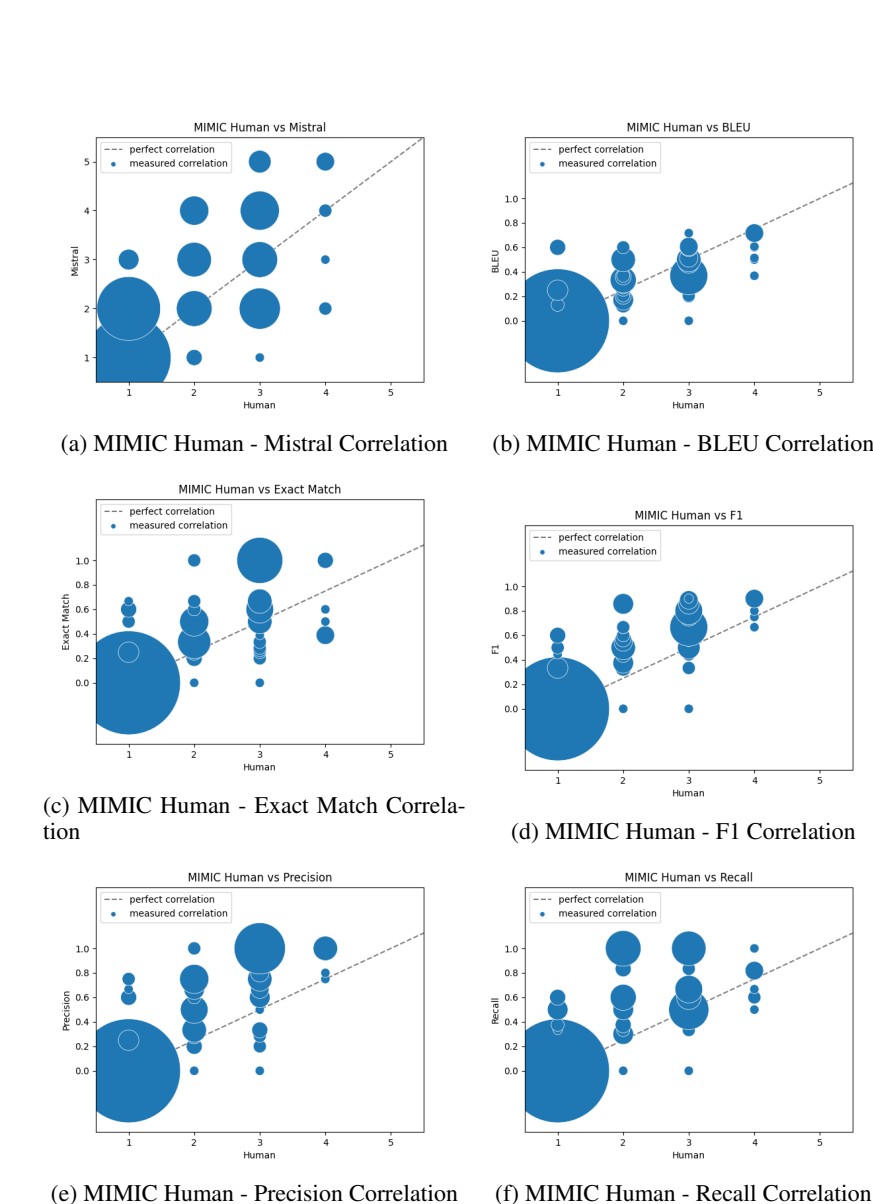

(a) MIMIC Human - Mistral Correlation

(b) MIMIC Human - BLEU Correlation

(c) MIMIC Human - Exact Match Correlation

(d) MIMIC Human - F1 Correlation

(e) MIMIC Human - Precision Correlation

(f) MIMIC Human - Recall Correlation

Figure 10: Scatter plots showing the correlation between the human ratings and respective other metrics on the MIMIC dataset. Size of the dots indicates the number of ratings that correspond to that point.

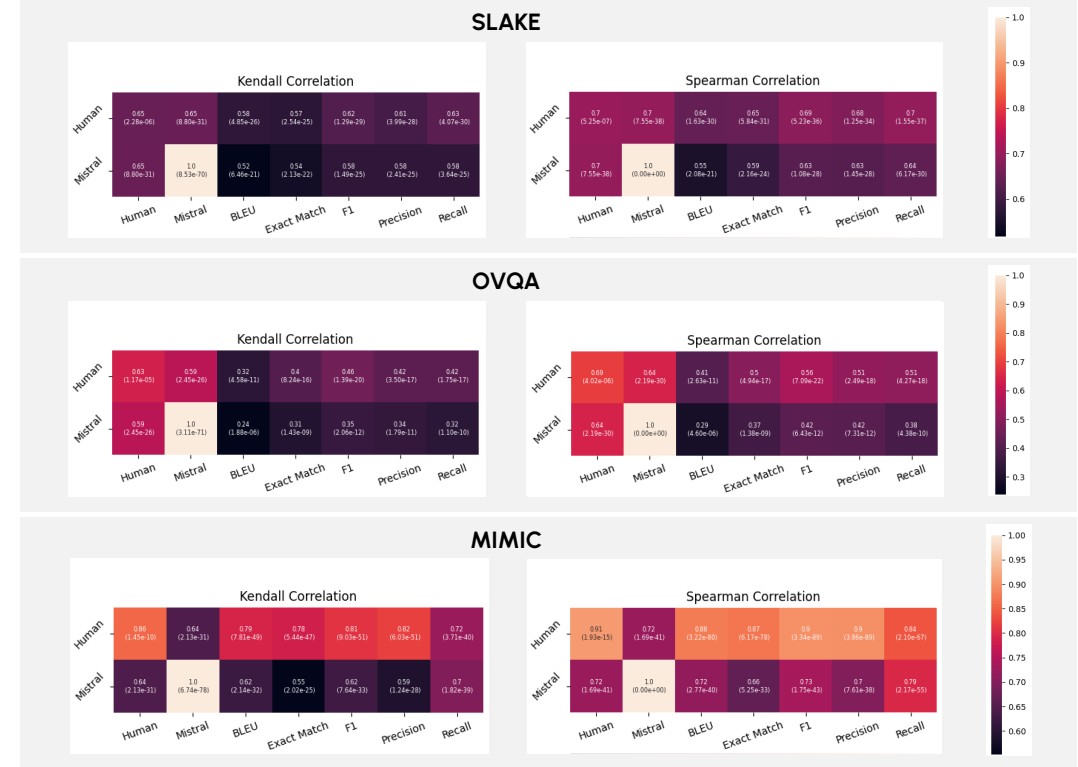

Figure 11: **Extended Results of the Human Rater Study**. Human interrater correlation is calculated between two human raters. Shown are the Kendall and Spearman correlation between the human rating and all traditional metrics as well as the correlation between Mistral and the traditional metrics.

## C    ROBUSTNESS STUDY DETAILS

### C.1    HYPERPARAMETER SEARCH

We performed several hyperparameter sweeps for each dataset and PEFT method in order to find suitable setups for the experiments in the PEFT robustness study. For the hyperparameter sweeps, we trained on the whole training set for each dataset and PEFT method and ran inference on the validation set. Training ran for 3 epochs and 3 seeds for each experiment.

### C.1.1 PROMPT TUNING

For prompt tuning, we performed the following hyperparameter sweeps:

- Number of tokens: $[40, 60, 80, 100]$
- Learning rate: $[3e - 2, 3e - 1]$

The results for SLAKE can be found in Table 1, for OVQA in Table 2, and for MIMIC in Table 3.

Table 1: Hyperparameter sweep for prompt tuning on the SLAKE dataset. Selected hyperparameters for the final PEFT robustness study are highlighted. Mean and standard deviation are reported for three seeds.

| # Tokens | Learning Rate | Closed-Ended (Mistral Accuracy) | Open-Ended (Mistral Score) |
|---|---|---|---|
| 40 | 3e-2 | 0.79 +/- 0.02 | 4.17 +/- 0.04 |
| 40 | 3e-1 | 0.8 +/- 0.02 | 4.19 +/- 0.05 |
| 60 | 3e-2 | 0.76 +/- 0.04 | 4.17 +/- 0.03 |
| 60 | 3e-1 | 0.81 +/- 0.01 | 4.17 +/- 0.05 |
| 80 | 3e-2 | 0.78 +/- 0.05 | 4.15 +/- 0.01 |
| 80 | 3e-1 | 0.82 +/- 0.01 | 4.18 +/- 0.02 |
| 100 | 3e-2 | 0.77 +/- 0.06 | 4.15 +/- 0.05 |
| 100 | 3e-1 | 0.81 +/- 0.0 | 4.18 +/- 0.03 |

Table 2: Hyperparameter sweep for prompt tuning on the OVQA dataset. Selected hyperparameters for the final PEFT robustness study are highlighted. Mean and standard deviation are reported for three seeds.

| # Tokens | Learning Rate | Closed-Ended (Mistral Accuracy) | Open-Ended (Mistral Score) |
|---|---|---|---|
| 40 | 3e-2 | 0.85 +/- 0.0 | 2.96 +/- 0.06 |
| 40 | 3e-1 | 0.85 +/- 0.01 | 2.95 +/- 0.05 |
| 60 | 3e-2 | 0.85 +/- 0.01 | 2.98 +/- 0.04 |
| 60 | 3e-1 | 0.85 +/- 0.0 | 2.97 +/- 0.04 |
| 80 | 3e-2 | 0.81 +/- 0.04 | 2.96 +/- 0.04 |
| 80 | 3e-1 | 0.84 +/- 0.01 | 2.99 +/- 0.02 |
| 100 | 3e-2 | 0.83 +/- 0.02 | 2.99 +/- 0.04 |
| 100 | 3e-1 | 0.85 +/- 0.0 | 3.0 +/- 0.03 |

Table 3: Hyperparameter sweep for prompt tuning on the MIMIC dataset. Selected hyperparameters for the final PEFT robustness study are highlighted. Mean and standard deviation are reported for three seeds.

| # Tokens | Learning Rate | Closed-Ended (Mistral Accuracy) | Open-Ended (Mistral Score) |
|---|---|---|---|
| 40 | 3e-2 | 0.67 +/- 0.02 | 3.17 +/- 0.03 |
| 40 | 3e-1 | 0.67 +/- 0.01 | 3.15 +/- 0.04 |
| 60 | 3e-2 | 0.68 +/- 0.01 | 3.14 +/- 0.01 |
| 60 | 3e-1 | 0.69 +/- 0.01 | 3.19 +/- 0.02 |
| 80 | 3e-2 | 0.66 +/- 0.05 | 3.17 +/- 0.01 |
| 80 | 3e-1 | 0.68 +/- 0.02 | 3.17 +/- 0.03 |
| 100 | 3e-2 | 0.68 +/- 0.02 | 3.19 +/- 0.03 |
| 100 | 3e-1 | 0.67 +/- 0.01 | 3.17 +/- 0.03 |

### C.1.2 LoRA

For LoRA, we performed the following hyperparameter sweeps:

- Rank: $[16, 32, 64, 128, 256]$
- Learning rate: $[3e - 5, 3e - 4]$

$\alpha$ is set to $2 \times$ Rank. The results for SLAKE can be found in Table 4, for OVQA in Table 5, and for MIMIC in Table 6. Note that some of the hyperparameter configurations led to instabilities during training loss, indicated by "NaN".

Table 4: Hyperparameter sweep for LoRA on the SLAKE dataset. Selected hyperparameters for the final PEFT robustness study are highlighted. Mean and standard deviation are reported for three seeds. Rows with "NaN" showed instabilities in the loss during training.

| Rank | Learning Rate | Closed-Ended (Mistral Accuracy) | Open-Ended (Mistral Score) |
|---|---|---|---|
| 16 | 3e-5 | 0.83 +/- 0.01 | 4.24 +/- 0.02 |
| 16 | 3e-4 | 0.82 +/- 0.01 | 4.23 +/- 0.05 |
| 32 | 3e-5 | 0.85 +/- 0.01 | 4.27 +/- 0.04 |
| 32 | 3e-4 | 0.73 +/- 0.07 | 4.2 +/- 0.03 |
| 64 | 3e-5 | 0.84 +/- 0.01 | 4.29 +/- 0.04 |
| 64 | 3e-4 | 0.52 +/- 0.07 | 3.01 +/- 1.28 |
| 128 | 3e-5 | 0.84 +/- 0.01 | 4.31 +/- 0.03 |
| 128 | 3e-4 | NaN | NaN |
| 256 | 3e-5 | 0.83 +/- 0.01 | 4.28 +/- 0.01 |
| 256 | 3e-4 | NaN | NaN |

Table 5: Hyperparameter sweep for LoRA on the OVQA dataset. Selected hyperparameters for the final PEFT robustness study are highlighted. Mean and standard deviation are reported for three seeds. Rows with "NaN" showed instabilities in the loss during training.

| Rank | Learning Rate | Closed-Ended (Mistral Accuracy) | Open-Ended (Mistral Score) |
|---|---|---|---|
| 16 | 3e-5 | 0.84 +/- 0.0 | 3.02 +/- 0.07 |
| 16 | 3e-4 | 0.83 +/- 0.02 | 3.08 +/- 0.03 |
| 32 | 3e-5 | 0.85 +/- 0.0 | 3.04 +/- 0.01 |
| 32 | 3e-4 | 0.82 +/- 0.01 | 2.99 +/- 0.04 |
| 64 | 3e-5 | 0.85 +/- 0.0 | 3.11 +/- 0.02 |
| 64 | 3e-4 | 0.65 +/- 0.0 | 2.04 +/- 0.1 |
| 128 | 3e-5 | 0.85 +/- 0.0 | 3.09 +/- 0.04 |
| 128 | 3e-4 | NaN | NaN |
| 256 | 3e-5 | 0.85 +/- 0.0 | 3.1 +/- 0.03 |
| 256 | 3e-4 | NaN | NaN |

Table 6: Hyperparameter sweep for LoRA on the MIMIC dataset. Selected hyperparameters for the final PEFT robustness study are highlighted. Mean and standard deviation are reported for three seeds. Rows with "NaN" showed instabilities in the loss during training.

| Rank | Learning Rate | Closed-Ended (Mistral Accuracy) | Open-Ended (Mistral Score) |
|---|---|---|---|
| 16 | 3e-5 | 0.7 +/- 0.01 | 3.31 +/- 0.01 |
| 16 | 3e-4 | 0.68 +/- 0.01 | 3.18 +/- 0.04 |
| 32 | 3e-5 | 0.71 +/- 0.0 | 3.33 +/- 0.02 |
| 32 | 3e-4 | 0.42 +/- 0.16 | 2.34 +/- 0.06 |
| 64 | 3e-5 | 0.71 +/- 0.01 | 3.33 +/- 0.03 |
| 64 | 3e-4 | NaN | NaN |
| 128 | 3e-5 | 0.7 +/- 0.0 | 3.35 +/- 0.04 |
| 128 | 3e-4 | NaN | NaN |
| 256 | 3e-5 | NaN | NaN |
| 256 | 3e-4 | NaN | NaN |

## C.2 $(IA)^3$

For $(IA)^3$, we performed the following hyperparameter sweeps:

- Learning rate: $[3e-3, 3e-2, 3e-1]$

The results for SLAKE can be found in Table 7, for OVQA in Table 8, and for MIMIC in Table 9.

Table 7: Hyperparameter sweep for $(IA)^3$ on the SLAKE dataset. Selected hyperparameters for the final PEFT robustness study are highlighted. Mean and standard deviation are reported for three seeds.

| Learning Rate | Closed-Ended (Mistral Accuracy) | Open-Ended (Mistral Score) |
|---|---|---|
| lr3e-3 | 0.63 +/- 0.02 | 3.74 +/- 0.02 |
| lr3e-2 | 0.83 +/- 0.01 | 4.28 +/- 0.02 |
| lr3e-1 | 0.65 +/- 0.01 | 4.21 +/- 0.05 |

Table 8: Hyperparameter sweep for $(IA)^3$ on the OVQA dataset. Selected hyperparameters for the final PEFT robustness study are highlighted. Mean and standard deviation are reported for three seeds.

| Learning Rate | Closed-Ended (Mistral Accuracy) | Open-Ended (Mistral Score) |
|---|---|---|
| lr3e-3 | 0.75 +/- 0.01 | 2.84 +/- 0.02 |
| lr3e-2 | 0.84 +/- 0.0 | 3.08 +/- 0.01 |
| lr3e-1 | 0.78 +/- 0.04 | 2.97 +/- 0.05 |

Table 9: Hyperparameter sweep for $(IA)^3$ on the MIMIC dataset. Selected hyperparameters for the final PEFT robustness study are highlighted. Mean and standard deviation are reported for three seeds.

| Learning Rate | Closed-Ended (Mistral Accuracy) | Open-Ended (Mistral Score) |
|---|---|---|
| lr3e-3 | 0.53 +/- 0.0 | 2.86 +/- 0.01 |
| lr3e-2 | 0.7 +/- 0.01 | 3.3 +/- 0.04 |
| lr3e-1 | 0.61 +/- 0.05 | 3.06 +/- 0.04 |

### C.3 DATASET DETAILS

#### C.3.1 SLAKE

The SLAKE dataset Liu et al. (2021a) is a bilingual radiological VQA dataset, containing English and Chinese questions. We use the English subset of the SLAKE dataset. The dataset is composed of MRI, CT, and X-ray images. All images are 2D, so for the MRI and CT images, single slices are extracted. For each question, metadata information about the location, the modality, and the content is provided. Overall, the images are split into 5 different body locations, 11 different content types (question types), and the mentioned three modalities.

The exact sizes of the dataset splits are listed in Table 10. Note that for the modality shift, we merged the test set with the OoD cases from the training set, since the images are distinct, and thus, the same image cannot appear in the training and test set. As this is not the case for the question type shift, we only use the OoD cases from the test set here.

Table 10: Size of the SLAKE dataset for the different splits.

| Split | i.i.d./OoD/all | # Cases |
|-------|----------------|---------|
| **Whole Dataset** | | |
| Train | all | 4866 |
| Validate | all | 1043 |
| **Modality Shift (OoD: X-Ray)** | | |
| Train | i.i.d. | 3448 |
| Test | i.i.d. | 689 |
| Test | OoD | 1779 |
| **Question Type Shift (OoD: Size)** | | |
| Train | i.i.d. | 4581 |
| Test | i.i.d. | 994 |
| Test | OoD | 56 |

#### C.3.2 OVQA

The OVQA dataset Huang et al. (2022) is an orthopedic VQA dataset, containing CT and X-Ray images. All images are 2D, so for the CT images, either a 3D rendering is shown as a 2D image or a single plane. For each question, metadata information is provided about the imaged organ (like the "location" in the SLAKE dataset), and the question type (like the "content" in SLAKE) is provided. The dataset contains 6 different question types and 4 different body parts.

The exact sizes of the dataset splits are listed in Table 11. We removed closed-ended questions with more than two categories to choose from and closed-ended questions where the categories to answer were not exactly contained in the question. As for the SLAKE dataset, we merged the questions from the training set to the OoD test set for the organ shift, but not for the question type shift.

Table 11: Size of the OVQA dataset for the different splits.

| Split | i.i.d./OoD/all | # Cases |
|-------|----------------|---------|
| **Whole Dataset** | | |
| Train | all | 13492 |
| Validate | all | 1645 |
| **Organ Shift (OoD: Leg)** | | |
| Train | i.i.d. | 8755 |
| Test | i.i.d. | 1044 |
| Test | OoD | 5350 |
| **Question Type Shift (OoD: Organ System)** | | |
| Train | i.i.d. | 11924 |
| Test | i.i.d. | 1420 |
| Test | OoD | 237 |

#### C.3.3 MIMIC-CXR-VQA

The MIMIC-CXR-VQA dataset Bae et al. (2023) is a chest X-ray dataset, which is built based on the MIMIC-CXR dataset Johnson et al. (2019), the MIMIC-IV dataset Johnson et al. (2023), and the Chest ImaGenome dataset Wu et al. (2021). For each question, the semantic type is specified. Three different semantic types are specified, which are "choose", "query", and "verify". For "choose", the

task is to choose between two options provided in the answer, but also both or none of the options can be correct. For "query", the task is to list all the categories that match the questions, e.g. all anatomical findings. Lastly, "verify" are yes/no questions. All the questions can be answered based on a fixed set of classes, where the dataset overall contains 110 answer labels. The answers are given as a list of the correct classes. We preprocess the questions differently, based on their semantic type: For the "choose" questions, whenever the list of answers contains both options, we change the answer to "both", and whenever the list of answers is empty, we change the answer to "none". For the "query" questions, we concatenate the list of answers to one string, with the answer labels being comma-separated. For the "verify" questions, we do not apply any specific preprocessing.

The information for the patient's gender, ethnicity, and age are taken from the MIMIC-IV dataset. Whenever the metadata information of a subject ID is not unique, we set it to "none". In the respective shifts, we exclude questions where the corresponding metadata field is not known, which includes all fields with "none", and for the ethnicity shift also the value "unknown/other". The exact sizes of the dataset splits are listed in Table 12.

Table 12: Size of the MIMIC dataset for the different splits.

| Split | i.i.d./OoD/all | # Cases |
|---|---|---|
| **Whole Dataset** | | |
| Train | all | 290031 |
| Validate | all | 73567 |
| **Gender Shift (OoD: Female)** | | |
| Train | i.i.d. | 147790 |
| Test | i.i.d. | 7277 |
| Test | OoD | 6120 |
| **Ethnicity Shift (OoD: Non-white)** | | |
| Train | i.i.d. | 171593 |
| Test | i.i.d. | 8101 |
| Test | OoD | 3713 |
| **Age Shift (OoD: Young)** | | |
| Train | i.i.d. | 155941 |
| Test | i.i.d. | 6686 |
| Test | OoD | 2076 |

### C.3.4 RATIO OF UNIQUE QUESTIONS IN THE DATASETS

Table 13: Ratio of unique questions in the datasets

| | | Overall | Unique | Ratio |
|---|---|---|---|---|
| **Train** | **MIMIC** | 290031 | 132387 | 0.46 |
| | **SLAKE** | 4866 | 579 | 0.12 |
| | **OVQA** | 13492 | 960 | 0.07 |
| **Val** | **MIMIC** | 73567 | 31148 | 0.42 |
| | **SLAKE** | 1043 | 314 | 0.3 |
| | **OVQA** | 1645 | 266 | 0.16 |
| **Test** | **MIMIC** | 13793 | 7565 | 0.55 |
| | **SLAKE** | 1050 | 313 | 0.3 |
| | **OVQA** | 1657 | 335 | 0.2 |

### C.4 DETAILED RESULTS OF THE ROBUSTNESS STUDY

Tables 14-19 show the detailed results of the robustness study. Further, Figure 12 shows the inter-method and inter-shift variability of the different PEFT methods, so not including full FT.

Table 14: **Robustness Results on the SLAKE Dataset.** Results with ± indicate the mean and standard deviation over three seeds. Note that the most frequent baseline can only be calculated for the i.i.d. set as for OoD too few questions match the training set. RR: Relative Robustness.

| | Modality Shift OoD: X-Ray | | | | | | Question Type Shift OoD: Size | | | | | |
| | Closed-Ended | | | Open-Ended | | | Closed-Ended | | | Open-Ended | | |
| | i.i.d. | OoD | RR | i.i.d. | OoD | RR | i.i.d. | OoD | RR | i.i.d. | OoD | RR |
|---|---|---|---|---|---|---|---|---|---|---|---|---|
| No Finetune | 0.59 | 0.29 | 0.49 | 2.91 | 2.79 | 0.96 | 0.54 | 0.47 | 0.87 | 2.86 | 3.16 | 1.11 |
| Full Finetune | 0.57 | 0.48 | 0.83 | 4.03 | 3.17 | 0.79 | 0.56 | 0.35 | 0.63 | 4.11 | 4.38 | 1.07 |
| Prompt | 0.85±0.01 | 0.62±0.03 | 0.73±0.05 | 4.22±0.04 | 3.69±0.06 | 0.87±0.02 | 0.85±0.01 | 0.49±0.12 | 0.57±0.14 | 4.17±0.01 | 4.35±0.16 | 1.04±0.04 |
| LoRA | 0.88±0.0 | 0.45±0.04 | 0.51±0.04 | 4.34±0.06 | 3.61±0.06 | 0.83±0.03 | 0.87±0.0 | 0.39±0.07 | 0.45±0.08 | 4.26±0.01 | 4.26±0.07 | 1.0±0.02 |
| $(IA)^3$ | 0.85±0.01 | 0.64±0.07 | 0.75±0.08 | 4.35±0.02 | 3.4±0.15 | 0.78±0.04 | 0.87±0.02 | 0.53±0.06 | 0.61±0.06 | 4.23±0.01 | 4.21±0.27 | 0.99±0.07 |
| Most Freq. | 0.69 | - | - | 3.22 | - | - | 0.696 | - | - | 3.05 | - | - |

Table 15: **No Image Baseline on the SLAKE Dataset.** Results with ± indicate the mean and standard deviation over three seeds. The model was trained with the same methods as Table 14 just without seeing the image content. RR: Relative Robustness.

| | Modality Shift OoD: X-Ray | | | | | | Question Type Shift OoD: Size | | | | | |
| | Closed-Ended | | | Open-Ended | | | Closed-Ended | | | Open-Ended | | |
| | i.i.d. | OoD | RR | i.i.d. | OoD | RR | i.i.d. | OoD | RR | i.i.d. | OoD | RR |
|---|---|---|---|---|---|---|---|---|---|---|---|---|
| No Finetune | 0.46 | 0.35 | 0.75 | 2.13 | 2.33 | 1.1 | 0.46 | 0.29 | 0.64 | 2.22 | 2.03 | 0.91 |
| Prompt | 0.55±0.01 | 0.5±0.01 | 0.91±0.03 | 3.24±0.0 | 1.88±0.06 | 0.58±0.02 | 0.59±0.05 | 0.55±0.07 | 0.94±0.18 | 3.18±0.04 | 2.26±0.6 | 0.71±0.19 |
| LoRA | 0.64±0.03 | 0.47±0.07 | 0.73±0.11 | 3.24±0.02 | 1.93±0.05 | 0.6±0.02 | 0.69±0.01 | 0.53±0.18 | 0.76±0.25 | 3.12±0.03 | 2.95±0.05 | 0.94±0.01 |
| $(IA)^3$ | 0.55±0.01 | 0.47±0.02 | 0.85±0.03 | 3.26±0.01 | 1.87±0.02 | 0.57±0.01 | 0.55±0.08 | 0.53±0.06 | 0.96±0.09 | 3.15±0.02 | 2.64±0.53 | 0.84±0.17 |

Table 16: **Robustness Results on the OVQA Dataset.** Results with ± indicate the mean and standard deviation over three seeds. Note that the most frequent baseline can only be calculated for the i.i.d. set as for OoD too few questions match the training set. RR: Relative Robustness.

| | Body Part Shift OoD: Leg | | | | | | Question Type Shift OoD: Organ System | | | | | |
| | Closed-Ended | | | Open-Ended | | | Closed-Ended | | | Open-Ended | | |
| | i.i.d. | OoD | RR | i.i.d. | OoD | RR | i.i.d. | OoD | RR | i.i.d. | OoD | RR |
|---|---|---|---|---|---|---|---|---|---|---|---|---|
| No Finetune | 0.42 | 0.4 | 0.96 | 2.4 | 2.45 | 1.02 | 0.43 | 0.33 | 0.75 | 2.39 | 1.94 | 0.81 |
| Full Finetune | 0.7 | 0.55 | 0.77 | 3.16 | 2.23 | 0.71 | 0.76 | 0.08 | 0.1 | 2.97 | 1.02 | 0.34 |
| Prompt | 0.86±0.0 | 0.75±0.01 | 0.87±0.01 | 3.12±0.02 | 2.38±0.02 | 0.76±0.01 | 0.82±0.04 | 0.86±0.01 | 1.05±0.06 | 2.9±0.01 | 1.7±0.02 | 0.59±0.01 |
| LoRA | 0.86±0.01 | 0.77±0.0 | 0.9±0.01 | 3.23±0.02 | 2.47±0.05 | 0.76±0.02 | 0.84±0.0 | 0.74±0.06 | 0.88±0.07 | 3.09±0.03 | 1.72±0.11 | 0.56±0.04 |
| $(IA)^3$ | 0.83±0.02 | 0.75±0.01 | 0.9±0.02 | 3.21±0.05 | 2.46±0.04 | 0.77±0.0 | 0.8±0.02 | 0.72±0.11 | 0.91±0.15 | 2.98±0.06 | 1.51±0.05 | 0.51±0.02 |
| Most Freq. | 0.75 | - | - | 2.57 | - | - | 0.73 | - | - | 2.23 | - | - |

Table 17: **No Image Baseline on the OVQA Dataset.** Results with ± indicate the mean and standard deviation over three seeds. The model was trained with the same methods as Table 16 just without seeing the image content. RR: Relative Robustness.

| | Body Part Shift OoD: Leg | | | | | | Question Type Shift OoD: Organ System | | | | | |
| | Closed-Ended | | | Open-Ended | | | Closed-Ended | | | Open-Ended | | |
| | i.i.d. | OoD | RR | i.i.d. | OoD | RR | i.i.d. | OoD | RR | i.i.d. | OoD | RR |
|---|---|---|---|---|---|---|---|---|---|---|---|---|
| No Finetune | 0.42 | 0.36 | 0.85 | 1.37 | 2.02 | 1.47 | 0.41 | 0.4 | 0.97 | 1.39 | 1.29 | 0.93 |
| Prompt | 0.74±0.01 | 0.69±0.02 | 0.93±0.01 | 2.63±0.1 | 2.12±0.07 | 0.81±0.01 | 0.67±0.03 | 0.44±0.01 | 0.66±0.02 | 2.35±0.06 | 1.26±0.07 | 0.54±0.02 |
| LoRA | 0.74±0.0 | 0.7±0.0 | 0.95±0.01 | 2.67±0.16 | 2.14±0.01 | 0.81±0.05 | 0.73±0.0 | 0.43±0.03 | 0.6±0.04 | 2.36±0.02 | 1.29±0.09 | 0.55±0.04 |
| $(IA)^3$ | 0.74±0.0 | 0.69±0.02 | 0.93±0.02 | 2.74±0.07 | 2.13±0.04 | 0.78±0.03 | 0.7±0.04 | 0.39±0.06 | 0.56±0.1 | 2.36±0.02 | 1.28±0.06 | 0.54±0.02 |

Table 18: **Robustness Results on the MIMIC Dataset.** Results with ± indicate the mean and standard deviation over three seeds. Note that the most frequent baseline can not be calculated as too few questions match the training set. RR: Relative Robustness.

| | Gender Shift OoD: Female | | | | | | Ethnicity Shift OoD: Non-white | | | | | | Age Shift OoD: Young | | | | | |
| | Closed-Ended | | | Open-Ended | | | Closed-Ended | | | Open-Ended | | | Closed-Ended | | | Open-Ended | | |
| | i.i.d. | OoD | RR | i.i.d. | OoD | RR | i.i.d. | OoD | RR | i.i.d. | OoD | RR | i.i.d. | OoD | RR | i.i.d. | OoD | RR |
|---|---|---|---|---|---|---|---|---|---|---|---|---|---|---|---|---|---|---|
| No Finetune | 0.51 | 0.49 | 0.97 | 2.36 | 2.36 | 1 | 0.5 | 0.49 | 0.98 | 2.37 | 2.34 | 0.99 | 0.51 | 0.47 | 0.92 | 2.36 | 2.36 | 1 |
| Full Finetune | 0.75 | 0.75 | 1.01 | 3.25 | 3.41 | 1.01 | 0.72 | 0.77 | 1.07 | 3.27 | 3.51 | 1.07 | 0.71 | 0.79 | 1.12 | 3.3 | 3.45 | 1.05 |
| Prompt | 0.52±0.06 | 0.54±0.05 | 1.04±0.03 | 3.25±0.05 | 3.3±0.05 | 1.01±0.01 | 0.54±0.14 | 0.64±0.14 | 1.19±0.01 | 3.14±0.11 | 3.37±0.28 | 1.07±0.05 | 0.51±0.14 | 0.66±0.07 | 1.37±0.35 | 3.19±0.03 | 3.35±0.08 | 1.05±0.02 |
| LoRA | 0.75±0.01 | 0.76±0.0 | 1.02±0.01 | 3.35±0.11 | 3.41±0.06 | 1.02±0.01 | 0.71±0.03 | 0.79±0.02 | 1.12±0.02 | 3.32±0.04 | 3.58±0.01 | 1.08±0.01 | 0.73±0.01 | 0.78±0.01 | 1.07±0.02 | 3.36±0.04 | 3.54±0.1 | 1.05±0.02 |
| $(IA)^2$ | 0.63±0.1 | 0.65±0.08 | 1.04±0.04 | 3.32±0.04 | 3.36±0.03 | 1.01±0.01 | 0.6±0.06 | 0.7±0.05 | 1.16±0.02 | 3.26±0.04 | 3.51±0.0 | 1.08±0.02 | 0.52±0.02 | 0.72±0.06 | 1.39±0.05 | 3.18±0.09 | 3.34±0.21 | 1.05±0.04 |

Table 19: **No Image Baseline on the MIMIC Dataset.** Results with ± indicate the mean and standard deviation over three seeds. The model was trained with the same methods as Table 18 just without seeing the image content. RR: Relative Robustness.

| | Gender Shift OoD: Female | | | | | | Ethnicity Shift OoD: Non-white | | | | | | Age Shift OoD: Young | | | | | |
| | Closed-Ended | | | Open-Ended | | | Closed-Ended | | | Open-Ended | | | Closed-Ended | | | Open-Ended | | |
| | i.i.d. | OoD | RR | i.i.d. | OoD | RR | i.i.d. | OoD | RR | i.i.d. | OoD | RR | i.i.d. | OoD | RR | i.i.d. | OoD | RR |
|---|---|---|---|---|---|---|---|---|---|---|---|---|---|---|---|---|---|---|
| No Finetune | 0.49 | 0.48 | 0.97 | 2.34 | 2.37 | 1.01 | 0.49 | 0.48 | 0.97 | 2.37 | 2.39 | 1.01 | 0.51 | 0.46 | 0.89 | 2.89 | 2.33 | 1.07 |
| Prompt | 0.53±0.0 | 0.5±0.0 | 0.95±0.0 | 2.71±0.01 | 2.63±0.01 | 0.97±0.0 | 0.54±0.0 | 0.42±0.0 | 0.78±0.0 | 2.75±0.02 | 2.43±0.02 | 0.88±0.01 | 0.6±0.03 | 0.35±0.02 | 0.6±0.04 | 2.93±0.04 | 2.25±0.02 | 0.77±0.01 |
| LoRA | 0.53±0.0 | 0.5±0.0 | 0.95±0.0 | 2.74±0.01 | 2.64±0.02 | 0.97±0.0 | 0.54±0.0 | 0.42±0.0 | 0.76±0.0 | 2.75±0.02 | 2.43±0.02 | 0.88±0.0 | 0.59±0.0 | 0.35±0.02 | 0.6±0.04 | 2.93±0.04 | 2.25±0.02 | 0.77±0.0 |
| $(IA)^2$ | 0.53±0.01 | 0.51±0.0 | 0.95±0.01 | 2.72±0.03 | 2.64±0.02 | 0.97±0.0 | 0.54±0.0 | 0.42±0.0 | 0.77±0.0 | 2.71±0.02 | 2.46±0.02 | 0.91±0.01 | 0.6±0.0 | 0.34±0.0 | 0.57±0.01 | 2.91±0.02 | 2.29±0.04 | 0.79±0.01 |

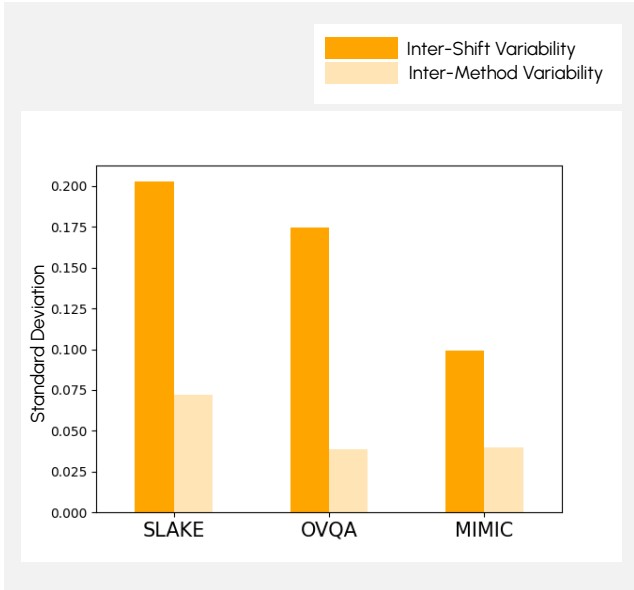

Figure 12: Standard deviation between shifts vs. standard deviation between PEFT methods, not including full FT. The type of shift has a higher impact on the robustness than the PEFT method.

## D    CORRUPTION STUDY

We compared the realistic shifts defined for our datasets (R1) with artificial shifts, meaning image corruptions, to assess whether artificial shifts correspond to real-world shifts. The artificial data shifts were generated through image corruptions, including blur, Gaussian noise, and brightness adjustments. They were applied in different strengths (low, medium, and high). We used OpenCV for the image corruptions with the settings shown in Table 20.

Table 20: Corruption settings for the artificial shifts. Brackets indicate the altered parameter for each corruption, [...,...] indicate ranges for the corruption where randomly a value in that range is chosen.

|  | Blur (Kernel Size) | Gaussian Noise (Mean) | Brightness (Alpha) |
| --- | --- | --- | --- |
| Low | 5 | [0, 0.06] | [1.1, 2] |
| Medium | 7 | [0.09, 0.15] | [2.5, 4] |
| High | 11 | [0.18, 0.25] | [4.5, 6] |

For this sample study, we used the LLaVA-Med model fine-tuned on the SLAKE dataset with the $(\texttt{IA})^3$ method. The i.i.d. and OoD samples for realistic shifts were as previously described (R1). For artificial shifts, the i.i.d. train and test samples were identical to those used for realistic shifts, while OoD test samples were created by corrupting the i.i.d. test images with varying strengths of blur, brightness, and noise. Each corruption method was applied with a probability of 0.5, with at least one corruption always being applied.

Table 21 shows the relative robustness results for both artificial and realistic shifts. The results show that both modality shift and question type shift exhibit lower relative robustness compared to all artificial shifts at low, medium, and high strengths. This suggests that artificial shifts, such as image corruption, fail to accurately represent the challenges posed by real-world, realistic shifts. The most prominent example here is the relative robustness of closed-ended questions under the question type shift (realistic shift), which is up to 96% compared to the realistic shift which only has 61%. The only exception where the realistic shift shows higher robustness is the question type shift on the open-ended questions, which is already nearly 100% on the realistic shift.

Table 21: Robustness results for the artificial and realistic shifts on SLAKE dataset

| | Corruption Shift (OoD: Corrupted i.i.d images) | | | | | | Corruption Shift (OoD: Corrupted i.i.d images) | | | | | |
| | Closed Ended | | | Open Ended | | | Closed Ended | | | Open Ended | | |
| | i.i.d. | OoD | RR | i.i.d. | OoD | RR | i.i.d. | OoD | RR | i.i.d. | OoD | RR |
| Low Corruption | 0.85±0.01 | 0.83±0.01 | 0.98±0.0 | 4.35±0.02 | 4.21±0.05 | 0.97±0.02 | 0.87±0.02 | 0.84±0.0 | 0.96±0.02 | 4.23±0.01 | 4.16±0.03 | 0.98±0.01 |
| Medium Corruption | 0.85±0.01 | 0.79±0.02 | 0.94±0.02 | 4.35±0.02 | 4.0±0.09 | 0.92±0.02 | 0.87±0.02 | 0.82±0.01 | 0.94±0.01 | 4.23±0.01 | 3.96±0.03 | 0.94±0.01 |
| High Corruption | 0.85±0.01 | 0.74±0.01 | 0.87±0.01 | 4.35±0.02 | 3.79±0.09 | 0.87±0.02 | 0.87±0.02 | 0.76±0.02 | 0.87±0.03 | 4.23±0.01 | 3.87±0.03 | 0.91±0.01 |

| | Modality shift (OoD: X-Ray) | | | | | | Question Type Shift (OoD: Size) | | | | | |
| | Closed Ended | | | Open Ended | | | Closed Ended | | | Open Ended | | |
| | i.i.d. | OoD | RR | i.i.d. | OoD | RR | i.i.d. | OoD | RR | i.i.d. | OoD | RR |
| Realistic Shift | 0.85±0.01 | 0.64±0.07 | 0.75±0.08 | 4.35±0.02 | 3.4±0.15 | 0.78±0.04 | 0.87±0.02 | 0.53±0.06 | 0.61±0.06 | 4.23±0.01 | 4.21±0.27 | 0.99±0.07 |

# E  MULTIMODAL SHIFTS

We conducted an ablation study on the OVQA dataset to evaluate the impact of a multimodal shift compared to the previously introduced unimodal shifts. This multimodal shift combines the Manifestation (Body Part) and Question Type Shifts reported in our experiments. Specifically, we defined the OoD set as samples featuring body part "Leg" and question type "Organ System", with all other samples classified as i.i.d. As shown in Figure 13, the multimodal shift demonstrates the lowest robustness compared to unimodal shifts, which is expected given that multimodal shifts represent a more extreme divergence than their unimodal components.

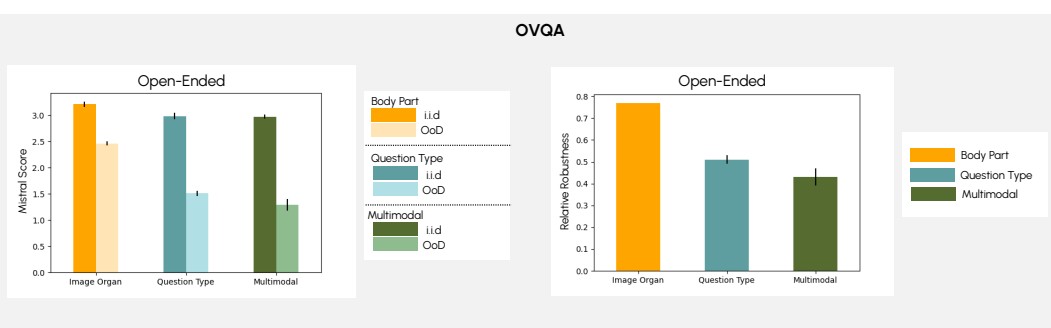

Figure 13: Performance results on OVQA dataset with image organ shift, question type shift and multimodal shift which combines image organ shift and question type shift.

