# OpenReview forum: "SURE-VQA: Systematic Understanding of Robustness Evaluation in Medical VQA Tasks"
_ICLR.cc/2025/Conference — Submitted to ICLR 2025_

### Official Review · Reviewer_UkoV · 2024-11-03

**Soundness:** 2
**Presentation:** 4
**Contribution:** 2
**Rating:** 6
**Confidence:** 4

**Summary:**

The robustness of vision-language models (VLMs) to distribution shifts is critical for safe deployment. However, existing benchmarks fail to provide realistic data shifts, rely on traditional metrics that do not capture the underlying semantics, and lack sanity baselines for interpreting performance. The paper proposes a flexible framework, called SURE-VQA, that addresses the above pitfalls in the evaluation of VLM robustness in the medical domain. The relevance of SURE-VQA is demonstrated via comparing the robustness of VLMs fine-tuned on medical domains. LLM-based metrics are also proposed and validated via a human rater study.

**Strengths:**

- The paper is well-written and easy to follow.

- A new dataset that consists of realistic shifts in both image and text domains is constructed. The dataset can be used to evaluate the robustness of various medical VLMs.

- The sanity baselines highlight the pitfall of VLMs in using linguistic shortcuts for predictions without the information in the images.

**Weaknesses:**

- The constructed dataset is a simple combination of three existing datasets. Although the paper aims to create more realistic shifts by considering both vision and text variations, in each VQA pair, only the shift in one modality is presented. Therefore, the SURE-VQA dataset looks incremental and does not provide novel insights into the robustness of VLMs against multimodal distribution shifts. It would be helpful to introduce shifts in both modalities in a VQA pair.

- The experiments in the paper only focus on one VLM. Results from other VLMs, such as Med-Flamingo [1] and PathChat [2],  may also be given to provide a more comprehensive view on the robustness of VLMs and to demonstrate the utility of the proposed dataset.

- It would be beneficial to introduce some mechanism to control the shifts in SURE-VQA to provide a more holistic view on the robustness of VLMs against distribution shifts. For example, control the number of shifted samples from each type in the test set.

- The use of large language models as evaluators does not appear to be very effective and may not be the best choice for the SURE-VQA dataset. First, on the SLAKE dataset, the correlation score achieved by the Mistral model is comparable to that of traditional metrics. Second, on the MIMIC dataset, traditional metrics yield higher correlation scores than the Mistral model. More importantly, as shown in Figure 3, the two human raters do not achieve much consensus on the SLAKE and OVQA datasets (with low correlation scores), making it unclear whether a high score by the Mistral model truly reflects the evaluation quality.

[1] Moor et al., Med-Flamingo: a Multimodal Medical Few-shot Learner, Machine Learning for Health Symposium, 2023.\
[2] Lu et al., A multimodal generative AI copilot for human pathology, Nature, 2024.

**Questions:**

- With multiple LLMs available, is there a specific reason for choosing the Mistral model as the evaluator?
- Why does the paper focus on analyzing different PEFT methods using the constructed dataset instead of different VLMs?

---

> ### Author Response · Authors · 2024-11-20
>
> Thank you again for your valuable comments and for taking the time to read our general reply, as well as considering our point-by-point comments here:
>
> ---
>
> W1. Although the paper aims to create more realistic shifts [...] only the shift in one modality is presented. Therefore, the SURE-VQA dataset [...] does not provide novel insights into the robustness of VLMs against multimodal distribution shifts [...]
> - We find this suggestion as an insightful addition to our paper. Therefore we have conducted an ablation study on OVQA dataset focusing on multimodal distribution shifts. This study is added to our Appendix E. Looking at the results, we can confirm the assumption that multimodal shift lowers robustness compared to unimodal shift since they are stronger.
>
> ---
>
> W2. The experiments in the paper only focus on one VLM [...]
> - We acknowledge this limitation and have addressed it in our manuscript. We focused on LLaVA-Med v1.5 since it is SOTA with highest community recognition (443 citations vs. 162 for Med-Flamingo). Expanding our study to multiple VLMs was infeasible due to the computational demands of extensive experiments (e.g. 189 experiments for hyperparameter tuning). Please refer to the general reply for more detailed discussion.
>
> ---
>
> W3. It would be beneficial to introduce some mechanism to control the shifts [...] For example, control the number of shifted samples from each type in the test set.
> - Thank you for your comment, in our study we create two test sets where one only contains i.i.d. data and the other one contains only OoD data. The reason why we used such a setting is because this approach aligns with existing literature, which often examines robustness in separate i.i.d. and OOD splits [1,2,3].
> - We would like to clarify that our study focuses on using realistic shifts rather than synthetic ones. While synthetic shifts allow for explicit control of severity, realistic shifts are more challenging to control for severity but better reflect real-world scenarios.
>
> [1] Shirnin et. al. "Analyzing the Robustness of Vision & Language Models"
> [2] Qiu et. al. "Benchmarking Robustness under Distribution Shift of Multimodal Image-Text Models"
> [3]  Chen et. al. "Benchmarking robustness of adaptation methods on pre-trained vision-language models".
>
> ---
>
> W4. The use of large language models as evaluators [...] may not be the best choice [...].
> - We have taken your comments into consideration and made several updates to our study to address your concerns.
> - To improve the robustness of our human evaluation, we increased the number of raters from two to five, resulting in a more reliable assessment of human consensus.
> - We observe that across all datasets, the human-human correlation is the highest, indicating strong alignment among the human raters compared to other metrics.
> - For SLAKE and OVQA, the correlation between human evaluations and Mistral is higher than that with traditional metrics, indicating Mistral's effectiveness as an evaluation tool. In contrast, for the MIMIC dataset, traditional metrics show higher correlation than Mistral due to MIMIC's structured format, where token matching works well. For detailed discussion please see concern 3 in our general reply.
>
> ---
>
> Q1. With multiple LLMs available, is there a specific reason for choosing the Mistral model as the evaluator?
> - Thank you for your question. We selected Mistral as our evaluator since at the time of our development, Mistral was among the state-of-the-art (SOTA) LLMs available in the open-source community, demonstrating significant improvements over models like Llama1 and Llama2 as stated in their paper [1] .
>
> [1] Jiang et. al. Mistral 7B
>
> ---
>
> Q2. Why does the paper focus on analyzing different PEFT methods using the constructed dataset instead of different VLMs?
> - Thank you for your question. It is important to note that our framework is flexible and not limited to PEFT methods alone. It can indeed be used to compare different VLMs in terms of their robustness.
> - Our focus on analysing different PEFT methods using the constructed dataset is motivated by the growing importance of efficient fine-tuning techniques for large models, especially in domains like the medical field with limited resources [1]. That’s why it’s both interesting and important to look into how the choice of PEFT strategy might affect a model’s robustness.
> - With this motivation, we see testing additional VLMs rather as an extension than a replacement, to our FT study. However, this extension is infeasible due to computational constraints. For details, please refer to our general reply.
>
> [1] Dutt et al. “Parameter-Efficient Fine-Tuning for Medical Image Analysis: The Missed Opportunity”
>
> ---
> Thank you once more for your constructive feedback. As we believe to have addressed your concerns, please let us know if there are any remaining concerns that would hinder a recommendation for acceptance.

---

> > ### Comment · Reviewer_UkoV · 2024-11-26
> >
> > Thanks for your detailed responses and additional experiments, which have addressed some of my concerns. Below are my additional comments.
> >
> > I think this dataset is more suitable to test fine-tuned models as it includes i.i.d. data for fine-tuning and o.o.d. dataset for testing. The experiments in the paper also focus on the fine-tuning setting. For testing pre-trained VLMs, I am not sure whether this would be a simple extension (response to Q2). Whether this dataset would be effective is still unknown, since there is no evidence to support this. More specifically, which portion of the dataset would you choose for testing pre-trained VLMs, i.i.d. data, o.o.d. data, or both? What if a pre-trained VLM has already seen the distribution?
> >
> > Even if the dataset is designed for testing fine-tuned models, it would still be beneficial to test some other VLMs to ensure the consistency of the conclusions. I acknowledge that this would have significantly increased the complexity and would be beyond your computational budget. But to support the utility of the proposed dataset, additional evidence is needed.
> >
> > (Minor) While the paper focuses on realistic shifts, the process of curating the dataset introduces certain assumptions on the shifts, which may not well align with those in the real world. Giving some flexibility in controlling the distribution of samples in the dataset might be helpful. For example, providing an option for selecting unimodal or multimodal shifts for evaluation, or varying the number of shifted samples.

---

> > > ### Author Response · Authors · 2024-11-27
> > >
> > > We sincerely thank you for taking the time to consider our initial response and provide your reply. Below, we provide our point-by-point answers:
> > >
> > > ---
> > >
> > > > I think this dataset is more suitable to test fine-tuned models [...] For testing pre-trained VLMs, I am not sure whether this would be a simple extension (response to Q2). [...] More specifically, which portion of the dataset would you choose for testing pre-trained VLMs, i.i.d. data, o.o.d. data, or both? What if a pre-trained VLM has already seen the distribution?
> > > - We agree that our current data shifts and definitions of i.i.d. and OoD are mainly relevant in the context of fine-tuning methods and not directly applicable to pre-trained VLMs. Consequently, as outlined in the response to Q2, we view extending to new VLMs as an addition rather than a replacement, given our currently selected shifts. For this, open-source VLMs that support fine-tuning can be used.
> > > - However, as discussed in lines 505-512 of our manuscript, our framework can generalize beyond the current setting by redefining i.i.d. and OoD relative to pre-training data. In such a setting, i.i.d. data must be drawn from the same distribution as the pretraining data, while the OoD data must come from a different distribution. A hypothetical example of a realistic distributional shift could involve using images from a new type of medical scanner (released after model training) as the OoD data, while using images from the same scanners known to be part of the pretraining data as the i.i.d. data. To clarify, when we refer to the generalizability of our framework, we mean the generalizability of the defined requirements R1-R3 rather than the specific dataset splits used in our fine-tuning study.
> > > - To summarize, there are two options for extensions in our framework: (1) retain the current shifts and incorporate new open-source VLMs that allow fine-tuning as base models, or (2) redefine i.i.d. and OoD shifts based on pre-training data, potentially introducing new datasets and shifts, while adhering to the requirements R1-R3.
> > >
> > > ---
> > >
> > > > Even if the dataset is designed for testing fine-tuned models, it would still be beneficial to test some other VLMs to ensure the consistency of the conclusions [...] But to support the utility of the proposed dataset, additional evidence is needed.
> > > - As mentioned in the previous point, we would like to emphasize that the main contribution of our paper lies not only in the dataset but in the proposed framework, which addresses pitfalls P1-P3 through its requirements R1-R3. These include:
> > >   - Selection of realistic data-inherent shifts
> > >   - Choice of appropriate metrics
> > >   - Implementation of essential "sanity baselines"
> > > - While, as acknowledged, adding more VLMs would extend the study, our findings from one VLM already provide valuable insights:
> > >   - Baselines that do not use the image information perform surprisingly well. Notably, the “Most frequent” baseline is not dependent on the VLM, making this finding generalizable across different VLMs and fine-tuning methods.
> > >   - The smaller the training sets, the more clearly PEFT methods outperform full FT
> > >   - No FT method clearly outperforms the others in terms of robustness
> > >   - Robustness trends are more consistent between methods than between shifts
> > >
> > > ---
> > >
> > > > (Minor) [...] Giving some flexibility in controlling the distribution of samples in the dataset might be helpful. For example, providing an option for selecting unimodal or multimodal shifts for evaluation, or varying the number of shifted samples.
> > > - While the results of our fine-tuning study showcase a range of shifts intended to cover a broad spectrum, the codebase supports the addition of new shifts based on the metadata available for each dataset. To emphasize this flexibility of our framework, we have updated the README of our codebase to include the possible configurations for the current datasets. Additionally, our multimodal ablation study demonstrates that the framework can be extended to include multimodal shifts.
> > > - Regarding the variation in the number of shifted samples, we believe there may be a misunderstanding. As mentioned in our previous response to W3., we create two separate test sets: one with only i.i.d. data and another with only OoD data, so there is no mixture of i.i.d. and OoD samples in the test set. We hope this clarifies that varying the number of shifted samples would not affect the evaluation setup. If this does not address your concern, could you elaborate on what you mean by “varying the number of shifted samples”?

---

> > > > ### Comment · Reviewer_UkoV · 2024-12-03
> > > >
> > > > Thank you for your detailed and thoughtful responses to my concerns.
> > > > Overall, I am satisfied with the response and appreciate the effort to incorporate flexibility and generalizability into your framework.

---

### Official Review · Reviewer_gM69 · 2024-11-04

**Soundness:** 3
**Presentation:** 2
**Contribution:** 2
**Rating:** 5
**Confidence:** 3

**Summary:**

"SURE-VQA" introduces a framework to evaluate the robustness of Vision-Language Models (VLMs) in medical Visual Question Answering (VQA), specifically in the context of real-world clinical shifts. The framework addresses three limitations in current robustness testing: reliance on synthetic shifts, lack of semantic-aware metrics, and absence of interpretability-oriented sanity baselines. The authors test various parameter-efficient fine-tuning (PEFT) methods on multiple medical datasets, comparing their robustness against distribution shifts such as gender, ethnicity, and age.

**Strengths:**

SURE-VQA’s focus on clinically relevant shifts (e.g., gender and ethnicity) is valuable for medical VQA applications, making it more practical for real-world use.
The code is open-sourced.

**Weaknesses:**

The authors present different types of shifts in the dataset in Figure 2 and this work is to evaluate the robustnees of VLMs in medical VQA tasks. So how do you consider the potential shifts between pre-training distribution in VLMs and dataset distribution? I mean, there may be a case that distribution shifts indeed exist in the dataset but the dataset distribution is totally included in the pre-training data of VLMs, then that makes no sense to evaluate the robustness of VLMs. I think a definition regarding 'distribution shifts' here is required [Han et al.] and hope authors can clarify this. Or have you performed any investigation to ensure the 'distribution shifts' are indeed exist among pre-training, training and test distributions?

Han, Z., Zhou, G., He, R., Wang, J., Wu, T., Yin, Y., Khan, S., Yao, L., Liu, T. and Zhang, K., 2023. How well does gpt-4v (ision) adapt to distribution shifts? a preliminary investigation. arXiv preprint arXiv:2312.07424.

**Questions:**

1. In human rater study, the kendall correlation between two human raters is supposed to be the highest and the Mistral is expected to be closest to humans' scores. But in figure 3, the correlation between humans is even the worst in SLAKE and not the highest in OVQA. How to understand this phenomenon?
2. In Figure 2, the authors define several distribution shifts. I was wondering if these shifts are commonly ackonwledged in OoD fields [Ye et al., Gulrajani et al.] or what's the basis for the authors to propopse these shift types?
3. As the paper focuses on medical setting, the selected shifts, though realistic, could be expanded to reflect more clinically meaningful scenarios. Shifts in patient demographics (e.g., age, gender, ethnicity) are useful here, but modality and especially question type shift may not correlated with medical setting.



Ye, N., Li, K., Bai, H., Yu, R., Hong, L., Zhou, F., Li, Z. and Zhu, J., 2022. Ood-bench: Quantifying and understanding two dimensions of out-of-distribution generalization. In Proceedings of the IEEE/CVF Conference on Computer Vision and Pattern Recognition (pp. 7947-7958).

Gulrajani, I. and Lopez-Paz, D., 2020. In search of lost domain generalization. arXiv preprint arXiv:2007.01434.

---

> ### Author Response · Authors · 2024-11-20
>
> Thank you again for your valuable comments and for taking the time to read our general reply, as well as considering our point-by-point comments here:
>
> ---
>
> W1. [...] There may be a case that distribution shifts indeed exist in the dataset but the dataset distribution is totally included in the pre-training data of VLMs [...]. I think a definition regarding 'distribution shifts' here is required [...]
> - Thank you for bringing up this important point. We think this is a misunderstanding because we totally agree with you: defining i.i.d. and OoD is challenging, particularly in the context of foundation models due to the extensive scope of training data, as also discussed in lines 508-511 of our manuscript (Generalizability of the Framework).
> - Thus, in our study, we specifically define i.i.d. versus OoD in terms of the distribution shift between the fine-tuning dataset and the test data, allowing us to control for the shifts in the data. Further, we ensured that the datasets used for fine-tuning and evaluation are sourced independently from those used during the pre-training, eliminating any potential data leakage.
> - To clarify our approach, we have updated the manuscript at lines 150-154 of our manuscript to more prominently highlight our definitions of i.i.d. and OoD within the scope of this study.
>
> ---
>
> Q1. In human rater study, the kendall correlation between two human raters is supposed to be the highest and the Mistral is expected to be closest to humans' scores. But in figure 3, the correlation between humans is even the worst in SLAKE and not the highest in OVQA. How to understand this phenomenon?
> - Thank you for highlighting this observation. Notably, with our more comprehensive human rater study (see general reply), this behaviour is no longer observable. Generally, we attributed this behavior to the fact that the metrics fall between the evaluations of the two human raters, resulting in a stronger correlation with the average human rating than between the individual raters themselves.
> - However, we would like to highlight once again that increasing the number of raters seems to reduce this effect, making the study design overall more robust.
>
> ---
>
> Q2. In Figure 2, the authors define several distribution shifts. I was wondering if these shifts are commonly ackonwledged in OoD fields [...]?
> - We mainly base our shifts on the taxonomy of Castro et al. [1], a highly cited work in the OoD field, with the taxonomy also being used in other papers, e.g. [2], [3], [4]. The only shift type that is not based on this taxonomy is the question type shift, which we introduce in order to have shifts in both modalities. This taxonomy is also positively perceived by reviewers 5eB8 and UkoV.
> - As also mentioned by Castro et al. “these terms correspond roughly to particular dataset shift scenarios studied in general ML literature, namely ‘covariate shift’, ‘concept shift’, ‘target shift’, ‘conditional shift’ and ‘domain shift’ [...].” [1]
> - Relating to [Gulranjani et al.], our shifts focus on “domains arising from natural processes”.
> - To highlight the relevance of our selected shifts, we have added citations to these works in lines 242-244 of our manuscript.
>
> [1] Castro et al. “Causality matters in medical imaging”
> [2] Bungert et al. “Understanding Silent Failures in Medical Image Classification”
> [3] Roschewitz et al. “Automatic correction of performance drift under acquisition shift in medical image classification”
> [4] Choi et al. “Translating AI to Clinical Practice: Overcoming Data Shift with Explainability”
>
> ---
>
> Q3. [...] The selected shifts, though realistic, could be expanded to reflect more clinically meaningful scenarios. Shifts in patient demographics [...] are useful here, but modality and especially question type shift may not correlated with medical setting.
> - We agree that there may be additional shifts worth investigating to further enhance clinical relevance and added it as a dimension to investigate in line 527 of our manuscript.
> - Regarding modality shift, we acknowledge that it is a rather extreme shift that may not directly occur in medical practice, but rather more subtle versions of it, like encountering new scanner types from different manufacturers. However, we included it to represent shifts of varying severities, allowing us to evaluate the model's robustness under more extreme conditions.
> - In contrast, we consider the question-type shift to be clinically relevant. It is plausible in a medical setting that a physician might ask about aspects not covered in the model's training or fine-tuning data. As medical knowledge and practices evolve over time, new question types may emerge that the model has not encountered before.
>
> ---
>
> Thank you once more for your constructive feedback. As we believe to have addressed your concerns, please let us know if there are any remaining concerns that would hinder a recommendation for acceptance.

---

### Official Review · Reviewer_5eB8 · 2024-11-04

**Soundness:** 2
**Presentation:** 3
**Contribution:** 3
**Rating:** 5
**Confidence:** 3

**Summary:**

This paper discusses a framework called SURE-VQA, which aims to systematically assess the robustness of visual-language models (VLMs) in medical visual question answering (VQA) tasks. The authors point out that while there are benchmarks for evaluating the robustness of VLMs, they fail to provide a sufficient framework to effectively address real-world issues, particularly concerning more realistic data variations. To address these problems, the authors propose the SURE-VQA framework, based on three key requirements: 1. Robustness should be evaluated against the inherent real-world shifts in VQA data; 2. Large language models (LLMs) should be used for more accurate semantic evaluations, replacing traditional label-matching metrics; 3. Meaningful baselines should be provided to assess the multimodal impacts of VLMs. The authors demonstrate the relevance of the SURE-VQA framework by studying the robustness of different parameter-efficient fine-tuning (PEFT) methods on three medical datasets.

**Strengths:**

1. The paper proposes a new evaluation framework (SURE-VQA) that addresses issues present in current evaluation methods for visual question answering (VQA) tasks, particularly in dealing with real-world distribution shifts. This framework may incorporate new evaluation metrics or methods, enhancing the evaluation standards for VQA models.

2. It utilizes abundant and representative datasets and proposes a comprehensive taxonomy that simulates four types of real-world data distribution shifts. The shifts encompass Acquisition, Question Type, Manifestation, and Population, broadly simulating various shifts that may occur in medical VQA datasets, achieving a comprehensive and reliable evaluation.

3. The study tests the impact of different PEFT algorithms on model robustness and includes comparisons with the sanity baseline, analyzing the model's dependence on visual information.

4. The figures are clear and significantly enhance the readability of the article.

**Weaknesses:**

1. The article briefly mentions the scope of robustness discussed in the Introduction and R1 sections, but it lacks a more explicit and comprehensive definition of the robustness of VLMs in Med VQA tasks. It may necessary to further compare with the existing work on the evaluation of VLM generalization ability to illustrate the innovation points.

2. Based on the descriptions in the article, the focus primarily lies on the interference resistance of VLMs against data distribution shifts. The authors list several scenarios of data distribution variations, which appear to resemble an evaluation of the model's domain adaptation capability. Are there any related works on VLM domain adaptation capabilities in the medical field that are comparable to this work?

3. Regarding the inconsistency in the model's robustness on open-ended questions versus closed-ended questions mentioned in sections 481-485 of "Comparison Between Shifts," where performance shows opposing results between SLAKE and OVQA, is there an explanation for this? Particularly in the SLAKE dataset, the performance on the Open-Ended Question Type Shift data appears to be higher on the out-of-distribution (OoD) dataset. Can this be elucidated?

4. The experimental results do not yield a unified conclusion or significant implications. It would be beneficial to summarize the key factors that significantly influence the robustness of VLMs in Med VQA tasks. Additionally, further discussion on methods for improving model robustness on out-of-distribution (OoD) data, based on the analysis of the impact of PEFT methods on model robustness, would be valuable. Relevant literature from general domains, such as "Ma J, Wang P, Kong D, et al. Robust visual question answering: Datasets, methods, and future challenges[J]. IEEE Transactions on Pattern Analysis and Machine Intelligence, 2024;" and "Yoon J S, Oh K, Shin Y, et al. Domain generalization for medical image analysis: A survey[J]. arXiv preprint arXiv:2310.08598, 2023," could provide useful insights.

**Questions:**

See weakness.

---

> ### Author Response · Authors · 2024-11-20
>
> Thank you again for your valuable comments and for taking the time to read our general reply, as well as considering our point-by-point comments here:
>
> ---
>
> W1. The article [...] lacks a more explicit and comprehensive definition of the robustness of VLMs in Med VQA tasks. It may necessary to further compare with the existing work on the evaluation of VLM generalization ability [...]
> - Thank you for your feedback. We have revised the manuscript to address your concern about providing a more comprehensive definition of the VLM robustness in Med VQA tasks. We have added an explanatory text to lines 043–045.
> - For a detailed discussion on existing VLM generalisation work, we kindly refer you to sections P1-P3 in the manuscript where we analyse the current literature and discuss the pitfalls.
>
> ---
>
> W2. [...] The authors list several scenarios [...], which appear to resemble an evaluation of the model's domain adaptation capability. Are there any related works on VLM domain adaptation capabilities in the medical field [...]?
> - Thank you for your comment. We would like to clarify that our study is more closely aligned with Domain/OoD generalisation rather than domain adaptation. While both address domain shifts, domain adaptation utilises target domain data to adapt models, whereas Domain/OoD generalisation works without access to target domain data and focuses on detecting shifts [1,2].
>    - "DA assumes the availability of labelled or unlabeled target data (i.e., unsupervised DA, UDA) for model adaptation" [1]
>    - "For OoD generalisation, one only has supervised data from the underlying data distribution on the training domain." [2]
> - We acknowledge the importance of highlighting this connection more explicitly. To address this, we have included a discussion on this topic in the revised manuscript at lines 043-045.
>
> [1] Yoon et al. "Domain Generalization for Medical Image Analysis: A survey"
> [2] Liu et al. "Learning Causal Semantic Representation for Out-of-Distribution Prediction"
>
> ---
>
> W3. Regarding the [...] the model's robustness [...] where performance shows opposing results between SLAKE and OVQA, is there an explanation for this? [...]
> - We appreciate the reviewer’s observation. The behaviour differences in SLAKE and OVQA stem from how OoD questions align with the training data. While OoD questions are absent from the training set, training questions can provide enough context for effective generalisation during testing. In SLAKE, open-ended questions like "What is the largest organ in the picture?" align with training data such as "What is the main organ in the image?", enabling better generalisation, whereas the closed-ended question "Which is smaller, [x] or [y]?" require reasoning not covered in training, leading to weaker performance.
> - In OVQA, the pattern is reversed, with the closed-ended question "Is this a study of [x]?" aligning with training tasks including questions like "Does this image show a normal [organ]?" and showing strong generalisation, while open-ended questions like "What bones are evaluated primarily?" require detailed reasoning not supported by the training data, resulting in poorer performance.
> - We acknowledge the importance of this point and have clarified it in the manuscript (lines 478-482).
>
> ---
>
> W4. [...] It would be beneficial to summarize the key factors that significantly influence the robustness of VLMs in Med VQA tasks. Additionally, further discussion on methods for improving model robustness on out-of-distribution (OoD) data, based on the analysis of the impact of PEFT methods on model robustness, would be valuable. Relevant literature from general domains [...] could provide useful insights.
> - We appreciate the reviewer’s insightful feedback and have updated our results section accordingly. Specifically, in lines 520–523, we now explicitly state that robustness alone is not a determining factor when selecting fine-tuning methods and different shifts may uniquely challenge model performance.
> - We also agree that the methods discussed by Yoon et. al. and Ma et. al. are relevant to enhance robustness, thus we cite them as a starting point of future work to enhance robustness and benchmark the methods within our framework in lines 533-535.
>
> ---
>
> Thank you once more for your constructive feedback. As we believe to have addressed your concerns, please let us know if there are any remaining concerns that would hinder a recommendation for acceptance.

---

### Official Review · Reviewer_aUke · 2024-11-05

**Soundness:** 3
**Presentation:** 3
**Contribution:** 2
**Rating:** 5
**Confidence:** 3

**Summary:**

This paper investigates the out-of-distribution (OOD) robustness of vision-language models (VLMs) in medical visual question answering (VQA) tasks. To improve the evaluation of model robustness, the authors introduce an evaluation framework, SURE-VQA, which encompasses three key requirements: Realistic Shifts, Appropriate Metrics, and Relevant Sanity Baselines. Experiments were conducted using parameter-efficient fine-tuning (PEFT) methods on three existing medical VQA datasets. The study concludes by highlighting three major shortcomings in current approaches and underscoring findings from the PEFT robustness analysis, which shows that no single PEFT method consistently outperforms the others in robustness.

**Strengths:**

1. The paper is well-structured and easy to follow, with clear figures and visualizations that enhance comprehension.
2. The three requirements proposed address key concerns in the field of OOD robustness.
3. The codebase is open-sourced, which may facilitate future researchers to evaluate other VLMs.

**Weaknesses:**

1. Only LLaVA-Med is evaluated in the experiments, which restricts the generalizability of the findings. Testing additional models, especially state-of-the-art general-purposed VLMs, would provide a more comprehensive view of robustness across different architectures.
2. The experimental section focuses heavily on PEFT methods, which may divert focus from the broader aim of evaluating overall robustness. This narrow focus could limit insights into robustness aspects that might arise from other fine-tuning or training approaches relevant to VLMs in medical VQA.
3. The primary contribution is the SURE-VQA framework itself. Although it establishes a valuable structure for evaluation, the paper lacks contributions in model development or methodological advancements that could directly improve robustness.
4. The paper uses Mistral and LLM-based metrics to evaluate semantic similarity, but it may not thoroughly address possible biases or limitations inherent in these evaluators, especially in the medical context.

**Questions:**

See the weaknesses.

---

> ### Author Response · Authors · 2024-11-20
>
> Thank you again for your valuable comments and for taking the time to read our general reply, as well as considering our point-by-point comments here:
>
> ---
>
> W1. Only LLaVA-Med is evaluated in the experiments [...].
> - We generally agree with your point and acknowledge that evaluating multiple VLMs could provide additional insights
> - However, as detailed in our general reply, the scope of our work focuses on proposing a general framework for evaluating VLM robustness, with the analysis of fine-tuning methods serving as an *exemplary use case* to demonstrate its utility
> - Extending the study to multiple VLMs would have exceeded the scope and computational budget but represents an important direction for future research
> - Please refer to our general reply for a more comprehensive discussion
>
> ---
>
> W2. The experimental section focuses heavily on PEFT methods [...]. This narrow focus could limit insights [...] that might arise from other fine-tuning or training approaches [...]
> - Thank you for your valuable suggestion regarding incorporating additional training schemes into our analysis. Based on your feedback, we have now included full fine-tuning as an additional training approach in our experiments
> - With these new results, we further underline the effectiveness of PEFT methods in the medical domain, which aligns with findings in [1]
> - We hope this extension to our analysis addresses your concern and believe it further strengthens the relevance of our findings. Please refer to our general reply and our updated manuscript for further details on the results of the newly added fine-tuning method.
>
> [1] Dutt et al. “Parameter-Efficient Fine-Tuning for Medical Image Analysis: The Missed Opportunity”
>
> ---
>
> W3. The primary contribution is the SURE-VQA framework itself. Although it establishes a valuable structure for evaluation, the paper lacks contributions in model development or methodological advancements [...]
> - We think there is a misunderstanding about the scope of ICLR. We specifically submitted to the primary area “datasets and benchmarks”, a dedicated area where the novelty does not lie in new methods.
> - The legitimacy is also evident in numerous ICLR papers sharing our format and type of contribution: E.g. Kahl et al. ICLR24 (oral), Jaeger et al. ICLR23 (Top 5%), Galil et al. ICLR23a (Top 25%), Galil et al. ICLR 2023b, Wiles et al. ICLR 2022 (oral), Zong et al. ICLR23 (Top 25%).
> - **Motivation**: The main observation we make is that currently there is a lack of literature highlighting key factors for evaluating robustness in the medical VQA task
> - **Novelty**: We are the first ones to
>     - Evaluate the robustness of (medical) VLMs on realistic data shifts inherent to the data
>     - Define key factors that should be adhered for a meaningful investigating robustness in VLMs
>     - Thereby, the focus of the aspect that influences the robustness can be varied by the concrete studies (in out case the focus was on fine-tuning methods)
> - **Concrete Impact:**
>     - Researchers can use our framework for
>         - benchmarking other components that might influence the robustness, e.g. the model architecture
>         - developing new methods that enhance the robustness
>     - Practitioners can use the findings of our benchmark as a guideline in method selection
>         - One concrete finding of our study is that no PEFT method consistently outperforms others in terms of robustness, and all are similarly robust, making the choice of PEFT method independent of robustness needs. We specifically highlight this in our updated manuscript in lines 520-522.
>
> ---
>
> W4. The paper uses Mistral [...] to evaluate semantic similarity, but it may not thoroughly address possible biases or limitations inherent in these evaluators [...].
> - We believe that a reliable metric should bring the predictor as close as possible to the gold standard, which, in the context of free-text evaluation are human raters [1,2,3]. Rather than analyzing potential biases of individual metrics, we demonstrate that the proposed metric aligns more closely with this gold standard, making it a preferable choice. We therefore leverage Mistral as an LLM evaluator, aiming to capture the semantic understanding that traditional metrics lack
> - Further, as outlined in our general reply, we made our human rater study now even more comprehensive by adding more human raters for a more robust evaluation of Mistral as an LLM-based metric
> - For a detailed discussion, please refer to the general reply
>
> [1] Celikyilmaz et al. “Evaluation of Text Generation: A Survey”
> [2] Yang et al. “Advancing Multimodal Medical Capabilities of Gemini”
> [3] Fu et al. “GPTScore: Evaluate as You Desire”
>
> ---
>
> Thank you once more for your constructive feedback. As we believe to have addressed your concerns, please let us know if there are any remaining concerns that would hinder a recommendation for acceptance.

---

### Author Response · Authors · 2024-11-20

We sincerely thank reviewers for their valuable comments. The reviewers generally agree on overall relevance to the community (“address key concerns in the field of OOD robustness”), the clear presentation in text and figures (“well-structured”, “figures are clear and significantly enhance the readability”), and the value for the community by providing the open-source codebase.
Next to the point-by-point responses, we will address three main concerns here:

---

**Concern 1: Only using one VLM in the evaluation** (raised by aUke, UkoV)
- We acknowledge that this is a limitation of our study, and we have explicitly highlighted this point in our manuscript as an area for future research.
- However, the focus of our paper is to propose a general framework for evaluating the robustness of VLMs which can be used for studying various aspects in future work, and our analysis of the fine-tuning methods provides an *exemplary use case* of the general framework
- Extending our study to multiple VLMs would have significantly increased the complexity and would be beyond our computational budget, especially given the extensive hyperparameter search that we performed for each PEFT method across multiple datasets (189 experiments for hyperparameters).
- We selected LLaVA-Med as it is the SOTA model with the most popularity in the community (e.g. 443 citations of LLaVA-Med vs. 162 of Med-Flamingo). Specifically, we used its most recent version (v1.5), ensuring up-to-date advancements and demonstrating the model's active maintenance and thus continued relevance in the field.

---

**Concern 2: Only focus on PEFT methods** (raised by aUke)
- As discussed in concern 1, we acknowledge that many other components in the VLM pipeline can be investigated in terms of their effect on robustness, and our paper aims at providing a foundational framework
- However, to make our study of fine-tuning (FT) more comprehensive, we added full FT as another FT method into our analysis.
- We updated the result figures and the analysis including full FT in our paper. In summary, key findings include
    - Full FT performs worse on the SLAKE and OVQA dataset and only partially performs on par or outperforms LoRA on the MIMIC dataset
    - Generally, the performance of full FT increases with increasing dataset size, confirming [1]
    - We observe one complete failure of full FT regarding the robustness on the OVQA question type shift. This also raises the Inter-Method variability (Fig. 6c).
    - Overall, since full FT does not outperform the PEFT methods in terms of performance or robustness, we conclude that PEFT methods are specifically suited for medical / low data scenarios [1].

[1] Dutt et al. “Parameter-Efficient Fine-Tuning for Medical Image Analysis: The Missed Opportunity”

---

**Concern 3: Mistral as Evaluator.** (raised by aUke, UkoV)

- We acknowledge that limiting our human rater study to two raters may raise concerns about generalizability, especially when interrater disagreement is high.
- To address this, we extended our human rater study to include five raters, enhancing its robustness and reliability. The updated results have been incorporated into our manuscript.
- As shown Fig. 3, human-human correlation is highest across all datasets, demonstrating strong alignment among raters compared to other metrics. Since human evaluations are considered the gold standard in the literature [1,2,3], our aim is to employ a metric that effectively captures semantic understanding and aligns closely with this standard. As seen in the examples in A.1, Fig. 7, traditional metrics lack this semantic understanding.
- For the SLAKE and OVQA datasets, the correlation to the human ratings is highest among the automated metrics, supporting its effectiveness as an evaluation tool.
- For the MIMIC dataset, as discussed in our paper (lines 288–295), traditional metrics exhibit higher correlations with the human than Mistral. This is likely due to the structured nature of MIMIC (comma separated classes as answer). In such cases, traditional metrics, which rely on exact token matching, perform well.

[1] Celikyilmaz et al. “Evaluation of Text Generation: A Survey”
[2] Yang et al. “Advancing Multimodal Medical Capabilities of Gemini”
[3] Fu et al. “GPTScore: Evaluate as You Desire”

---

We have thoroughly revised our manuscript to address the provided feedback and uploaded the revision, where we highlight the modified parts in red. Changes include:
- Adding full FT as additional FT method
- Enhancing the human rater study by having 5 human raters
- Doing an ablation with multimodal shifts, meaning shifts in the image and text content (requested by reviewer UkoV)
- Clarifications of points raised in individual reviews, please see our point-by-point answers for details

We believe that these updates address the stated concerns. Please find our point-by-point answers in the respective reviewer sections.

---

### Author Response · Authors · 2024-11-25
**Follow-up**

Dear reviewers,
as the discussion period is nearing its end, we wanted to kindly check if you have any remaining points for us to address. We have responded to all concerns raised, in addition to running necessary experiments to strengthen our manuscript. If we do not receive any further comments on our rebuttal, we assume that all points of criticism have been addressed appropriately.
We appreciate your valuable feedback and are happy to clarify anything further if needed.

---

### Meta-Review · Area_Chair_HkA4 · 2024-12-20

**Metareview:**

The paper under review proposes the SURE-VQA framework to assess the robustness of VLMs in medical VQA. It has strengths like a well-structured presentation and open-source codebase. However, weaknesses include limited model evaluation (only LLaVA-Med) and a narrow focus on PEFT methods. Reviewers had concerns about the lack of a comprehensive robustness definition, inconsistent model performance across datasets, and the choice of evaluators. The authors responded with explanations, additional experiments (such as including full FT and enhancing the human rater study), and clarifications. Overall, while efforts have been made to address concerns, the paper still has areas that need further consideration, especially regarding the generalization of results and the comprehensiveness of the evaluation, which suggests a reject.

**Additional Comments On Reviewer Discussion:**

During the rebuttal period, reviewers raised points like limited model evaluation (only one VLM used), narrow focus on certain methods (e.g., PEFT), lack of clear robustness definition, concerns about evaluators, and issues with datasets. Authors addressed these by explaining scope and budget limitations, adding additional methods, enhancing human rater studies, clarifying definitions, and conducting relevant studies. In weighing the points, while authors' responses mitigated some concerns, areas like generalization of results and comprehensiveness of evaluation still needed further consideration for the final decision.

---

### Decision · Program_Chairs · 2025-01-22

Reject